# Structured Reinforcement Learning
# for Combinatorial Decision-Making

**Heiko Hoppe[1]**   **Léo Baty[2]**   **Louis Bouvier[2]**   **Axel Parmentier[2]**   **Maximilian Schiffer[1]**

[1]Technical University of Munich    [2]École des Ponts

{heiko.hoppe,schiffer}@tum.de
{leo.baty,louis.bouvier,axel.parmentier}@enpc.fr

## Abstract

Reinforcement learning (RL) is increasingly applied to real-world problems involving complex and structured decisions, such as routing, scheduling, and assortment planning. These settings challenge standard RL algorithms, which struggle to scale, generalize, and exploit structure in the presence of combinatorial action spaces. We propose *Structured Reinforcement Learning* (SRL), a novel actor-critic paradigm that embeds combinatorial optimization-layers into the actor neural network. We enable end-to-end learning of the actor via Fenchel-Young losses and provide a geometric interpretation of SRL as a primal-dual algorithm in the dual of the moment polytope. Across six environments with exogenous and endogenous uncertainty, SRL matches or surpasses the performance of unstructured RL and imitation learning on static tasks and improves over these baselines by up to 92% on dynamic problems, with improved stability and convergence speed.[1]

## 1 Introduction

Reinforcement learning has achieved remarkable progress during the last decade, expanding beyond its early success stories of Atari games and robotic control. Recently, increasing attention has been given to real-world industrial problems, such as vehicle routing, inventory planning, machine scheduling, and assortment optimization [e.g., Nazari et al., 2018, Kool et al., 2019, Hottung and Tierney, 2022]. Unlike traditional RL applications, these industrial problems often involve large-scale combinatorial decision-making, which challenges classical RL algorithms [cf., Hildebrandt et al., 2023]. In particular, existing algorithms struggle with: i) the exponential size of the action spaces, which renders action selection computationally intractable and hinders efficient exploration; and ii) leveraging the combinatorial structure of the action spaces using standard neural architectures, often resulting in poor generalization and unstable learning dynamics [cf., Yuan et al., 2022]. From a methodological perspective, most industrial problems translate into combinatorial Markov Decision Processes (MDPs), i.e., MDPs with combinatorial action spaces, that remain the focus of this paper.

**Problem 1** (Combinatorial Markov Decision Process). *We consider MDPs with states $s \in \mathcal{S}$, actions $a \in \mathcal{A}(s) \subset \mathbb{R}^{d(s)}$, rewards $r$, and transition probabilities $\mathbb{P}(s', r \mid s, a)$ with next state $s'$. In combinatorial MDPs, $\mathcal{A}(s)$ is the set of feasible solutions of a combinatorial problem. We denote its convex hull as the moment polytope $\mathcal{C}(s) := \mathrm{conv}(\mathcal{A}(s))$. As in stochastic optimization [Bertsekas and Shreve, 1996], we use a latent noise variable $\xi \in \Xi$ with probability $p(\xi \mid s, a)$, such that the transition to $(s', r)$ given $(s, a, \xi)$ is deterministic. We distinguish exogenous noise, where the distribution of $\xi$ does not depend on $a$, and endogenous noises, where the distribution of $\xi$ depends on $a$.*

---

[1]Our code is available at `https://github.com/tumBAIS/Structured-RL`.

*Given unknown $\mathbb{P}(s', r|s, a)$, we aim to find the reward-maximizing policy*

$$\bar{\pi} \in \arg\max_{\pi} \mathbb{E}_{\pi, \mathbb{P}} \left[ \sum_{t=0}^{T} \gamma^t r_t \right],$$

*over $[0, T]$, given a discount factor $\gamma$. We further define Q-values, satisfying the Bellman equation*

$$Q^{\pi}(s_t, a_t) = \mathbb{E}_{\mathbb{P}}[r_t] + \gamma \, \mathbb{E}_{\mathbb{P}} \left[ \max_{\tilde{a}_{t+1}} Q^{\pi}(s_{t+1}, \tilde{a}_{t+1}) \right],$$

*using expectations with respect to $\mathbb{P}(s', r \mid s, a)$.*

The limitations of standard RL algorithms in solving such Combinatorial MDPs (C-MDPs) render them insufficient to solve many industrial problems. To overcome these drawbacks, we propose *Structured Reinforcement Learning* (SRL) – a novel actor-critic RL paradigm that embeds combinatorial optimization (CO)-layers into neural actors, enabling to exploit the underlying problem structure.

**State of the Art** Several research communities have studied learning and decision-making over structured spaces, including structured prediction [e.g., Nowozin and Lampert, 2011], differentiable optimization layers [e.g., Amos and Kolter, 2017], and hierarchical reinforcement learning [e.g., Bacon et al., 2017]. While these works provide tools for embedding structure, they do not directly tackle end-to-end reinforcement learning in environments with combinatorial action spaces.

Prior approaches to solving C-MDPs include handcrafted decision rules [e.g., Liyanage and Shanthikumar, 2005, Huber et al., 2019] and predict-and-optimize algorithms [e.g., Alonso-Mora et al., 2017, Bertsimas and Kallus, 2020]. While the former fail to model complex dynamics or constraints, the latter typically rely on imitation learning and separate prediction from optimization, which impairs performance in dynamic settings [cf., Enders et al., 2023]. In contrast, CO-augmented Machine Learning (COAML)-pipelines integrate combinatorial optimization directly into model architectures, allowing end-to-end learning [e.g., Parmentier, 2022, Dalle et al., 2022]. Here, the key challenge is to differentiate through the CO-layer. Existing approaches include problem-specific relaxation schemes [e.g., Vlastelica et al., 2020], and more general strategies, e.g., using Fenchel-Young losses [e.g., Blondel et al., 2020, Berthet et al., 2020]. The latter allow continuous training over discrete structures and are increasingly used for differentiating COAML-pipelines [e.g., Dalle et al., 2022]. In C-MDPs, prior work either uses offline expert imitation [e.g., Baty et al., 2024, Jungel et al., 2024] or treats the CO-layer as an action mask in unstructured RL [e.g., Hoppe et al., 2024, Woywood et al., 2025]. The former requires access to expert solutions, while the latter often leads to unstable gradients. Imitation approaches also suffer from insufficient exploration in multi-stage problems.

SRL builds on classical policy-gradient algorithms, e.g., REINFORCE, PPO, and SAC [cf., Williams, 1992, Haarnoja et al., 2018], which perform well in conventional MDPs but are challenged by the size and structure of C-MDP action spaces [Hildebrandt et al., 2023]. Problem-specific neural network designs often lack generalizability across applications and struggle in dynamic contexts [e.g., Bello et al., 2017, Dai et al., 2017], while neural improvement methods rely on existing initial solutions [e.g., Yuan et al., 2022, Hottung and Tierney, 2022]. Value-based methods often assume decomposable critics [e.g., Xu et al., 2018, Liang et al., 2022], limiting applicability in structured domains. Alternative approaches transform a task-specific Q-network into a mixed-integer linear program, which potentially increases solution time and limits the range of usable network architectures [e.g., Xu et al., 2025]. SRL is also related to offline RL, which frequently relies on imitation learning [e.g., Figueiredo Prudencio et al., 2024], but differs by operating online and updating from on-policy targets.

**Contribution** To address the challenges outlined above, we propose a novel reinforcement learning paradigm for solving C-MDPs by integrating CO-layers into actor-critic architectures. Specifically, we introduce *Structured Reinforcement Learning* (SRL), a new framework for the end-to-end training of COAML-pipelines using only collected experience. SRL replaces the neural actor with a combinatorial policy defined by a score-generating network and a CO-layer. To enable end-to-end learning despite the non-differentiability of the CO-layer, SRL combines stochastic perturbation and Fenchel-Young losses to construct smooth actor updates, enabling stable policy improvement. We further provide a geometric analysis that interprets SRL as a sampling-based primal-dual method in the dual of the moment polytope, connecting structured learning and RL from a theoretical lens.

We demonstrate the effectiveness of SRL across six representative environments, including both static and dynamic decision problems with exogenous and endogenous uncertainty. On static problems, SRL improves by up to 54% on unstructured deep reinforcement learning baselines, and matches the performance of Structured Imitation Learning (SIL), despite requiring no expert supervision. On dynamic problems, SRL consistently outperforms SIL by up to 78% and unstructured deep reinforcement learning baselines (i.e., PPO) by up to 92%, while exhibiting lower variance and faster convergence across all settings.

## 2 Methodology

In the following, we introduce *Structured Reinforcement Learning*, a novel actor-critic framework tailored to environments with combinatorial action spaces. The main rationale of SRL is the embedding of a CO-layer in the actor architecture, which turns the actor from a plain neural network into a COAML-pipeline. This allows us to map high-dimensional states to score vectors via neural networks, while leveraging combinatorial optimization to determine the best actions with respect to the score vectors. Establishing this algorithmic paradigm while ensuring end-to-end learning for the new actor requires additional changes, specifically i) identifying a loss function that allows to differentiate through the CO-layer when learning by experience, and ii) rethinking the policy evaluation scheme.

In the following, we first detail the foundations of the resulting new algorithmic paradigm. Afterward, we provide a geometric analysis that formalizes our learning scheme, which can be interpreted as a sampling-based primal-dual algorithm.

### 2.1 Structured Reinforcement Learning

Figure 1 sketches the elements of our SRL agent, which extends the standard actor-critic paradigm by integrating a CO-layer into the actor, replacing the conventional neural network policy representation. The black elements illustrate the actor pipeline during inference: a neural network maps the current state $s$ to a score vector $\theta$, which is then passed into the CO-layer $f$. This CO-layer selects a feasible action $a \in \mathcal{A}(s)$ by solving a combinatorial problem, ensuring that selected actions are both scalable in the dimension of $\mathcal{A}(s)$ and valid with respect to the problem domain. The blue elements illustrate the architecture used during training: we sample perturbed versions of the score vector to generate candidate actions via the CO-layer. We then evaluate these actions using a critic network, and utilize a softmax-based aggregation to choose a *target action* $\widehat{a}$ for the actor update. We train the actor end-to-end by minimizing a *Fenchel-Young loss* between $\theta$ and $\widehat{a}$, which enables end-to-end backpropagation. The following details each of our algorithmic components.

**Combinatorial Actor** Consider a C-MDP as detailed in Problem 1, where $\mathcal{A}(s)$ denotes the feasible action set in state $s$. Deep RL algorithms typically use neural networks to represent the policy as a distribution over the action space. In combinatorial settings, a neural network may struggle to encode a distribution on the exponentially-large action space $\mathcal{A}(s)$. To overcome this challenge, we adopt a CO-augmented Machine Learning-pipeline as the actor architecture. As illustrated in Figure 1, this architecture combines a statistical model $\varphi_w$ with a combinatorial optimizer $f$. The statistical model $\varphi_w$, usually a neural network parameterized by weights $w$, observes contextual information

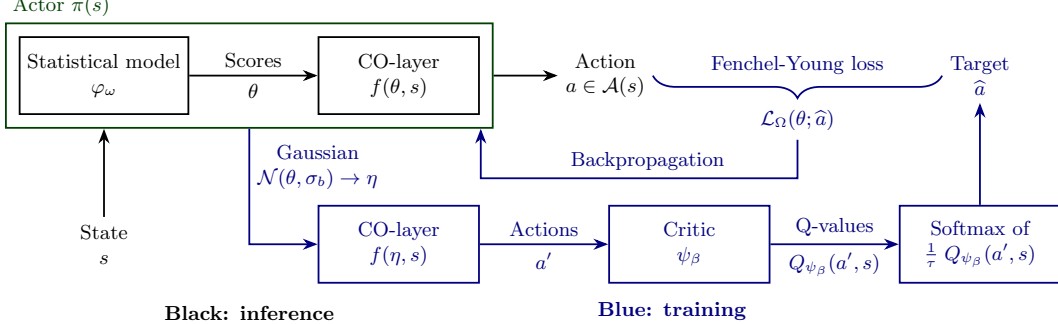

Figure 1: Overview of the Structured Reinforcement Learning algorithm.

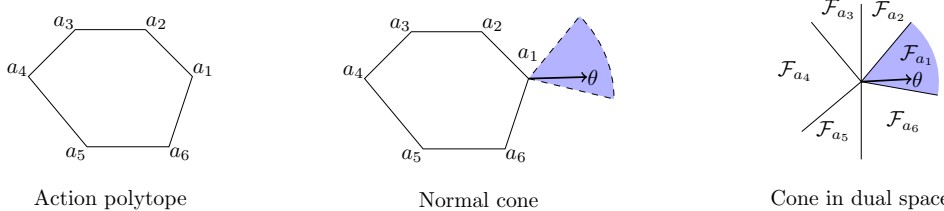

Figure 2: Left: action polytope $\mathcal{C}(s) = \text{conv}\left(\mathcal{A}(s)\right)$. Middle: normal cone for which $f(\theta, s) = a_1$, right: normal cone $\mathcal{F}_{a_1}$ in dual space.

provided by the C-MDP state $s$. Using this information, the model estimates a latent score vector $\theta$ with dimension $d$. The combinatorial optimizer $f$ uses $\theta$ as coefficients of a linear objective function and generates an action $a$ by solving the CO problem

$$f(\theta, s) : a = \operatorname*{argmax}_{\tilde{a} \in \mathcal{A}(s)} \langle \theta | \tilde{a} \rangle$$

given problem-specific constraints that define the feasible action set $\mathcal{A}(s)$. Formally, the resulting combinatorial actor reads $f(\varphi_w(s), s)$, such that we define the respective actor policy as $\pi_w(\cdot|s) := \delta_{f(\varphi_w(s),s)}$. Intuitively, one can interpret this policy as a Dirac distribution on the output of $f$.

In the COAML-pipeline, the statistical model $\varphi_w$ encodes contextual information, which enables generalizing across states, capturing variable dependencies, and anticipating C-MDP dynamics. The CO-layer $f$ enforces combinatorial feasibility, facilitates a structured exploration of the action space, and improves scalability by mapping score vectors $\theta$ to potentially high-dimensional action spaces $\mathcal{A}(s)$. COAML-pipelines have been used successfully to address combinatorial problems using imitation learning, highlighting their suitability for C-MDPs [e.g., Parmentier, 2022, Baty et al., 2024]. For a detailed introduction to COAML-pipelines and a motivating example, we refer to Appendix A.

**End-to-end actor learning**   Training the actor parameters $w$ using gradient-based methods poses significant challenges. The CO-layer $f$ is piecewise constant with respect to the action space, resulting in uninformative (i.e., zero) derivatives almost everywhere. Geometrically, this behavior can be understood by considering the convex hull of the action space, $\mathcal{C}(s) = \text{conv}(\mathcal{A}(s))$, which forms a polytope as depicted in Figure 2. The score vector $\theta$ determines the direction of the objective function, and each vertex of $\mathcal{C}(s)$ corresponds to a *normal cone*, defined as the set of all $\theta$ mapped to the same vertex $a$ by $f$. These cones partition the dual space of $\mathcal{C}(s)$, forming the *normal fan* of $\mathcal{C}(s)$. When $\theta$ crosses a cone boundary, the mapping $f$ exhibits an abrupt change, assigning $\theta$ to a different action, thereby illustrating the piecewise constant nature of $f(\theta, s)$. Since $f(\theta, s)$ deterministically maps scores $\theta$ to actions $a$, it can be viewed as an action post-processing step within the environment. Under this perspective, one may interpret $\theta$ as the action space and parameterize a distribution over $\theta$ using $\varphi_w$, enabling the use of standard RL policy gradient methods via score-function estimators [Mohamed et al., 2020]. However, treating $f$ as part of the environment effectively induces a piecewise constant and highly non-smooth reward function, which exacerbates gradient variance and leads to substantial difficulties in practice, e.g., by deteriorating or prohibiting convergence.

To address these limitations, we propose *Structured Reinforcement Learning*, a primal-dual RL algorithm that employs Fenchel-Young losses to update the actor. Fenchel-Young losses define a surrogate objective that is convex in the output of the statistical model and allows for smooth gradient propagation. Differentiating this surrogate objective reduces to solving a convex optimization problem via stochastic gradient descent. We estimate these gradients using a pathwise estimator, which is known for its low variance [Blondel and Roulet, 2024].

Following the workflow illustrated in Figure 1, we outline SRL in Algorithm 1: After collecting experience and sampling transitions from the replay buffer, we perturb the score vector $\theta$ using a Gaussian distribution to generate a set of perturbed vectors $\eta$. We pass each $\eta$ through the CO-layer to obtain candidate actions $a'$, which are evaluated by the critic. We then compute a softmax-weighted target action $\widehat{a}$ based on their Q-values. Finally, we update the actor by minimizing the Fenchel-Young loss between $\theta$ and $\widehat{a}$; and update the critic using standard temporal-difference errors.

---

**Algorithm 1** Structured Reinforcement Learning

---

**Initialize** actor with model $\varphi_w$, critic $\psi_\beta$ and target critic $\psi_{\overline{\beta}}$ networks
**for** $e$ episodes **do**
    **Generate** trajectories, store and sample transitions $j$
    **for** $j$ transitions **do**
        **Perturb** $\theta_j = \varphi_w(s_j)$ using $Z \sim N(\theta_j, \sigma_b)$, sample $m$ $\eta_j$, solve $f(\eta_j, s_j)$ for each $\eta_j$
        **Calculate** target action $\widehat{a}_j = \left( \text{softmax}_{a'_j} \frac{1}{\tau} Q_{\psi_\beta}(s_j, a'_j) \right)$
        **Update** actor using $\mathcal{L}_\Omega(\theta; \widehat{a})$                          ▷ using a second perturbation
        **Update** critic by one step of gradient descent using $J(\psi_\beta) = \left( Q_{\psi_\beta}(s_j, a_j) - y_j \right)^2$
    **end for**
**end for**

---

Originally proposed for imitation learning [Blondel et al., 2020, Berthet et al., 2020], the Fenchel-Young loss has become an established loss function for the end-to-end training of COAML-pipelines [e.g., Parmentier and T'Kindt, 2023, Baty et al., 2024]. We adopt it in SRL since it is convex in the output of the statistical model $\theta = \varphi_w(s)$ and differentiable with respect to the latter, while leveraging the structure of the CO-layer $f$. The Fenchel-Young loss $\mathcal{L}_\Omega(\theta; \widehat{a})$ compares the difference between the objective values of the estimated action $a$ under the parameterization $\theta$ and the target action $\widehat{a}$

$$\mathcal{L}_\Omega(\theta; \widehat{a}) = \max_{a \in \mathcal{A}(s)} \theta^\top a - \theta^\top \widehat{a}. \tag{1}$$

We aim to find a statistical model $\varphi_w$ that predicts $\theta$ to minimize the Fenchel-Young loss $\min_\theta \mathcal{L}_\Omega(\theta; \widehat{a})$. Due to the piecewise constant CO-layer $f$, the loss in this form is neither smooth or convex in $\theta$. To address this problem, we introduce a Gaussian perturbation $Z \sim \mathbb{N}(0, \varepsilon)$ such that the loss reads

$$\mathcal{L}_\Omega(\theta; \widehat{a}) = \mathbb{E}\left[ \max_{a \in \mathcal{A}(s)} (\theta + Z)^\top a \right] - \theta^\top \widehat{a}. \tag{2}$$

Using an alternative definition, given a regularization function $\Omega : \mathbb{R}^d \to \mathbb{R} \cup \{+\infty\}$ and its Fenchel conjugate $\Omega^*$, the Fenchel-Young loss $\mathcal{L}_\Omega(\theta; \widehat{a})$ generated by $\Omega$ is defined over $\text{dom}(\Omega^*) \times \text{dom}(\Omega)$ as

$$\mathcal{L}_\Omega(\theta; \widehat{a}) := \Omega^*(\theta) + \Omega(\widehat{a}) - \langle\theta|\widehat{a}\rangle = \sup_{a \in \text{dom}(\Omega)} \left( \langle\theta|a\rangle - \Omega(a) \right) - \left( \langle\theta|\widehat{a}\rangle - \Omega(\widehat{a}) \right). \tag{3}$$

For a given $\theta \in \text{dom}(\Omega^*)$, we introduce the regularized prediction as $\sup_{a \in \text{dom}(\Omega)} \langle\theta|a\rangle - \Omega(a)$. The Fenchel-Young loss measures the non-optimality of $a \in \text{dom}(\Omega)$ as a solution of the regularized prediction problem. It is nonnegative and convex in $\theta$. If in addition $\Omega$ is proper, convex, and lower semi-continuous, $\mathcal{L}_\Omega$ reaches zero if and only if $a$ is a solution of the regularized prediction problem.

The Fenchel-Young loss requires a target action $\widehat{a}$, which SRL estimates online without relying on offline expert demonstrations. As visualized in Figure 3, SRL explores the action space around the deterministic action $a$ using a perturbation, and leverages the critic to compute a softmax-weighted local target action $\widehat{a}$. To construct $\widehat{a}$, SRL perturbs the score vector $\theta$ using a Gaussian distribution $Z \sim N(\theta, \sigma_b)$ with standard deviation $\sigma_b$, and samples $m$ perturbed scores $\eta$. Each $\eta$ yields a candidate action $a' = f(\eta, s)$, which is evaluated by the critic $\psi_\beta$. We then compute

$$\widehat{a} = \text{softmax}_{a'} \left( \frac{1}{\tau} Q_{\psi_\beta} \right) = \sum_{a'} a' \frac{\exp\left( \frac{1}{\tau} \cdot Q_{\psi_\beta}(s, a') \right)}{\sum_{a'} \exp\left( \frac{1}{\tau} \cdot Q_{\psi_\beta}(s, a') \right)}, \tag{4}$$

with temperature parameter $\tau$, controlling the sharpness of the softmax. If we decrease $\tau$, $\widehat{a}$ approaches the greedy action $\text{argmax}_{a'} \left( Q_{\psi_\beta}(s, a') \right)$. If we increase $\tau$, $\widehat{a}$ approaches the uniform average $\frac{1}{m} \sum_{a'} a'$. Note that $\widehat{a} \in \mathcal{A}(s)$ does not have to hold, as it is not executed in the environment, but only serves as a training signal for the Fenchel-Young loss.

The softmax-based estimator for $\widehat{a}$ provides key benefits in structured action spaces. First, it promotes exploration by distributing credit across multiple actions. Second, it reduces critic overestimation bias via value averaging [cf., van Hasselt, 2010, Fujimoto et al., 2018]. Third, it avoids selecting edge actions, thereby enhancing stability and preventing premature convergence to suboptimal policies.

To ensure sufficient exploration of the state space $\mathcal{S}$, we introduce stochasticity into the forward pass during training: we perturb the score vector $\theta$ using Gaussian noise $Z \sim \mathcal{N}(\theta, \sigma_f)$ with exploration

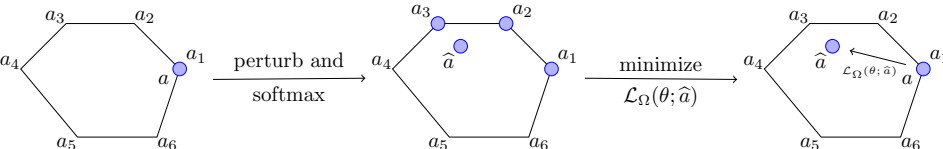

Figure 3: Schematic representation of SRL update step: unperturbed action $a$ (left), perturbed actions and target action $\widehat{a}$ (middle), Fenchel-Young loss $\mathcal{L}_\Omega(\theta; \widehat{a})$ (right).

standard deviation $\sigma_f$, sample one perturbed vector $\eta$, and select an action $a = f(\eta, s)$. Overall, we use three Gaussian perturbations of $\theta$, distinguished by their standard deviations: $\sigma_f$ to facilitate exploration of $\mathcal{S}$ in the forward pass, $\sigma_b$ to ensure exploration of the combinatorial action space $\mathcal{A}(s)$ during target action selection, and $\varepsilon$ to regularize the CO-layer $f$ for the Fenchel-Young loss.

**Critic architecture and stability**    SRL employs a critic network $\psi_\beta$, parameterized by weights $\beta$, which estimates Q-values $Q^\pi(s, a) = Q_{\psi_\beta}(s, a)$. We update the critic using standard temporal-difference (TD) learning: given observed transitions $(s_t, a_t, r_t, s_{t+1})$, the critic minimizes the TD loss $(Q_{\psi_\beta}(s_t, a_t) - y_t)^2$, with target value $y_t = r_t + \gamma Q_{\psi_{\bar\beta}}(s_{t+1}, a_{t+1})$. As common in RL, we update the target network weights $\overline{\beta}$ slowly to stabilize training, i.e., setting $\psi_{\overline{\beta}} \leftarrow \psi_\beta$ at the end of each episode. To mitigate the critic's overestimation bias, which is especially prevalent in large combinatorial spaces, we adopt double Q-learning techniques in complicated environments [cf., van Hasselt, 2010, Fujimoto et al., 2018], computing Q-values using the average of two critics.

While SRL follows a standard actor-critic architecture, combinatorial actions introduce unique challenges. Notably, SRL does not require the Q-function to be decomposable in the same dimension as the score vector $\theta$, unlike methods relying on linear or factored critics. This flexibility supports complex environments, e.g., industrial settings, but prevents direct optimization over actions. In particular, computing $\operatorname{argmax}_a Q_{\psi_\beta}(s, a)$ is generally infeasible for large combinatorial $\mathcal{A}(s)$, as it requires solving a hard optimization problem for each evaluation during training. This complexity increase further precludes the direct use of primal-dual techniques [cf., Bouvier et al., 2025], which require a critic-dependent optimization over actions. Instead, SRL adopts a sampling-based approach, which enables a tractable estimate of the best update direction, without requiring explicit optimization over the critic.

## 2.2   Geometrical insights

We focus our geometrical discussion on the static case ($T = 1$) of contextual stochastic combinatorial optimization. While this may appear as a simplification compared to the general multi-stage case ($T > 1$), it serves two purposes. First, it allows us to establish direct connections to the well-studied class of contextual stochastic combinatorial optimization problems [Sadana et al., 2025] and recent findings on primal-dual optimization schemes in this context [Bouvier et al., 2025]. Second, when learning a critic function via RL, the multi-stage decision problem effectively reduces to a single-stage problem with respect to the critic, since the critic approximates the expected cumulative return from any given state or state-action pair. Studying the static case thus not only clarifies the structure of our problem but also aligns with how value functions are typically learned and analyzed.

In this setting, we introduce a latent noise variable [Bertsekas and Shreve, 1996] $\xi \in \Xi$, with conditional distribution $p(\xi \mid s, a)$, such that the transition to $(s', r)$ given $(s, a, \xi)$ is deterministic. We distinguish between *exogenous noise*, where the distribution of $\xi$ is independent of the action $a$, and *endogenous noise*, where $p(\xi \mid s, a)$ explicitly depends on $a$. The objective of finding an optimal policy can then be formulated as

$$\bar\pi \in \operatorname*{argmax}_\pi \mathbb{E}_{\pi, \mathbb{P}}\big[r(s, a, \xi)\big]. \tag{5}$$

In this context, Bouvier et al. [2025] introduced a primal-dual algorithm for empirical risk minimization, making connections with mirror descent ([Nemirovsky et al., 1983, Bubeck, 2015]). This algorithm has nice theoretical and practical properties. However, it relies on the following: i) A combinatorial optimizer to solve $\max_{a \in \mathcal{A}(s)} \langle \theta | a \rangle$ for any $\theta \in \mathbb{R}^{d(s)}$. ii) A reward $r$ based on an exogenous noise variable $\xi$. This random variable $\xi$ is not observed when choosing the action $a \in \mathcal{A}(s)$,

but a posteriori. iii) A combinatorial optimizer to solve $\max_{a \in \mathcal{A}(s)} r(s, a, \xi) + \langle \theta | a \rangle$ when $\xi$ is observed.

In an RL setting, point i) typically holds, but points ii) and iii) do not stand. Indeed, the transition probability $\mathbb{P}$ is general, and the noise may be endogenous and not observed at all. Besides, the reward function is typically a black-box. One could think of replacing the black-box objective function of Equation (5) by a learned critic $\max_\pi \mathbb{E}_\pi \big[ Q_{\psi_\beta}(s, a) \big]$. However, in combinatorial optimization, the non-linearity of the critic function makes its optimization with respect to $a$ intractable. The key idea of Algorithm 1 is to sample a few atoms $(a_i)_{i \in [m]}$, and compute an expectation (softmax) involving a critic function $Q_{\psi_\beta}$. In Proposition 2 below, we highlight that in the static case, and with a fixed critic, this approach can be seen as a primal-dual algorithm, leveraging an additional sampling step. We show that it leads to tractable updates in our RL setting. To formalize this, we need to introduce a few definitions and background on optimization over the distribution simplex.

**Policies as solutions of regularized optimization over the distribution simplex** For a given state $s \in \mathcal{S}$, let $\Delta^{\mathcal{A}(s)}$ be the probability simplex over $\mathcal{A}(s)$. We recall that $\mathcal{C}(s)$ is the convex hull of the action space, also called moment polytope. Let $A(s) = (a)_{a \in \mathcal{A}(s)}$ be the wide matrix having one column per action. We introduce $\Omega_{\Delta^{\mathcal{A}(s)}} : \Delta^{\mathcal{A}(s)} \to \mathbb{R} \cup \{+\infty\}$ a regularization function, such that its restriction to the affine hull of $\Delta^{\mathcal{A}(s)}$ is Legendre-type [Rockafellar, 1970]. From this regularization over the simplex, we define a regularization over the moment polytope $\mathcal{C}(s)$ as follows. Let $\mu \in \mathcal{C}(s)$, $\Omega_{\mathcal{C}(s)}(\mu) := \min_{q \in \Delta^{\mathcal{A}(s)} : A(s)q = \mu} \Omega_{\Delta^{\mathcal{A}(s)}}(q)$. Every function $c(\cdot)$ on the combinatorial (exponentially large but finite) space $\mathcal{A}(s)$ can be seen as a long vector $\gamma = \big( c(a) \big)_a \in \mathbb{R}^{\mathcal{A}(s)}$. The distribution simplex $\Delta^{\mathcal{A}(s)}$ is the dual of this score space, and we can use $\Omega_{\Delta^{\mathcal{A}(s)}}$ to create mappings between them. More precisely, a policy maps a state $s \in \mathcal{S}$ to a distribution over the corresponding combinatorial action set $q \in \Delta^{\mathcal{A}(s)}$. To define such policies, we map a state $s$ to a direction vector $\theta = \varphi_w(s)$; then lift it to the score space $\gamma_\theta = A(s)^\top \theta = (\langle \theta | a \rangle)_{a \in \mathcal{A}(s)}$; and finally to a distribution $q = \nabla \Omega^*_{\Delta^{\mathcal{A}(s)}}(\gamma_\theta)$. Our regularized actor policy parameterized by $w$ is defined as

$$\pi_w(\cdot|s) = \underset{q \in \Delta^{\mathcal{A}(s)}}{\operatorname{argmax}} \{ \langle \underbrace{A(s)^\top \overbrace{\varphi_w(s)}^{\theta \in \mathbb{R}^{d(s)}}}_{\gamma_\theta \in \mathbb{R}^{\mathcal{A}(s)}} | q \rangle - \Omega_{\Delta^{\mathcal{A}(s)}}(q) \} = \nabla \Omega^*_{\Delta^{\mathcal{A}(s)}} \big( A(s)^\top \varphi_w(s) \big). \quad (6)$$

In Equation (6), we use the results of convex duality to write the argmax as a gradient of the Fenchel conjugate of $\Omega_{\Delta^{\mathcal{A}(s)}}$. The learning problem is to find the $w$ that maximizes $\mathbb{E}_{\pi_w, \mathbb{P}} \big[ r(s, a, \xi) \big]$. In practice, during inference, we do not regularize (see Section 2.1), thus obtaining a Dirac policy.

**A sampling-based primal-dual algorithm for the actor update** We consider a fixed state $s \in \mathcal{S}$ leading to a fixed action space $\mathcal{A}$, and thus drop the latter from the notation of spaces and regularization functions, and omit the neural network from the policy defined in Equation (6). Given a fixed critic function $Q_{\psi_\beta}$, we introduce the score vector $\gamma_\beta = \big( Q_{\psi_\beta}(a) \big)_{a \in \mathcal{A}} \in \mathbb{R}^{\mathcal{A}}$. Bouvier et al. [2025] introduce the following algorithm and show its convergence in a restricted setting.

$$\mu^{(t+1)} = A \nabla \Omega^*_\Delta \big( A^\top \theta^{(t)} + \frac{1}{\tau} \gamma_\beta \big), \quad (7a)$$

$$\theta^{(t+1)} \in \partial \Omega_{\mathcal{C}}(\mu^{(t+1)}). \quad (7b)$$

Here, $\Omega^*$ is the Fenchel-conjugate of $\Omega$, $\nabla \Omega^*$ the gradient of $\Omega^*$, and $\partial \Omega$ the sub-differential of $\Omega$. The following proposition, proved in Appendix B, shows that the static version of Algorithm 1 can be seen as a variant of the primal-dual algorithm presented in Bouvier et al. [2025], enhancing it with an additional sampling step.

**Proposition 2.** *The actor update in the static version of Algorithm 1 can be written as*

$$(a_i^{(t+\frac{1}{2})})_{i\in[m]} \sim_{i.d.} \nabla\Omega_{\varepsilon,\Delta}^*(A^\top\theta^{(t)}), \tag{8a}$$

$$\hat{q}_m^{(t+\frac{1}{2})} = \frac{1}{m}\sum_{i=1}^m \delta_{a_i^{(t+\frac{1}{2})}}, \tag{8b}$$

$$\gamma_m^{(t+\frac{1}{2})} \in \partial\Omega_\Delta(\hat{q}_m^{(t+\frac{1}{2})}), \tag{8c}$$

$$\mu^{(t+1)} = A\nabla\Omega_\Delta^*\big(\gamma_m^{(t+\frac{1}{2})} + \frac{1}{\tau}\gamma_\beta\big), \tag{8d}$$

$$\theta^{(t+1)} \in \partial\Omega_{\varepsilon,\mathcal{C}}(\mu^{(t+1)}), \tag{8e}$$

*where $\delta_a$ is the Dirac distribution on $a$, $\Omega_\Delta$ is the negentropy, and $\Omega_{\varepsilon,\Delta}$ is the conjugate of the sparse perturbation, both detailed in Appendix B. Since $\hat{q}_m^{(t+\frac{1}{2})}$ is sparse by design, we discuss the definition of gradients and sub-gradients at the (relative) boundary of the domains in Appendix B.*

In Equations (8), for convenience in the implementation, we involve two distinct regularization functions on the distribution simplex $\Delta^{\mathcal{A}}$, detailed in Appendix B. The main difference between Equations (7) and Equations (8) is the sampling step for the primal update. Recall that involving the critic in Equation (7a) may be intractable. Indeed, the critic function may be highly nonlinear, and the resulting nonlinear combinatorial optimization problem may be intractable. Instead, Equation (8) is based on a sampling step, which only requires $m$ evaluations of the critic, which is tractable even when optimizing it with respect to $a \in \mathcal{A}(s)$ is not. In both primal-dual algorithms, the dual update (of $\theta$) is equivalent to solving a convex optimization problem, precisely minimizing a Fenchel-Young loss generated by $\Omega_{\mathcal{C}}$. It is very convenient in practice, since the weights $w$ of our actor policy can be updated via stochastic gradient descent, although it relies on a piecewise constant CO-layer $f$.

## 3  Numerical studies

**Studied environments**   We evaluate SRL across six environments that reflect industrial applications with large combinatorial action spaces. Appendix C describes the experimental setup, and Appendix D details the environments. We first consider three static environments – common industrial benchmarks from Dalle et al. [2022] – namely, a Warcraft Shortest Paths Problem, a Single Machine Scheduling Problem, and a Stochastic Vehicle Scheduling Problem. As discussed in Appendix E, SRL matches the performance of SIL, while relying solely on access to a (black-box) cost function and without requiring the expert knowledge necessary for SIL. These results underline the versatility and broader potential of SRL, even though we designed it primarily for dynamic settings. We now discuss our findings for the dynamic environments in more detail.

The dynamic environments model online decision-making in C-MDPs. We consider: i) a Dynamic Vehicle Scheduling Problem (DVSP), based on the Dynamic Vehicle Routing Problem introduced by Kool et al. [2022], Baty et al. [2024]. This problem with exogenous uncertainty requires serving spatio-temporally distributed requests that are revealed over time. The goal is to find cost-minimizing routes while fulfilling all requests. ii) A Dynamic Assortment Problem (DAP), adapted from Dulac-Arnold et al. [2016] andChen et al. [2020]. This problem with endogenous uncertainty involves selecting item assortments that are shown to customers, whose choices follow a multinomial logit model. Item features evolve based on past decisions, the goal is overall revenue maximization. iii) A Gridworld Shortest Paths Problem (GSPP), inspired by gridworld and robotic control tasks [Chandak et al., 2019, Zhang et al., 2020]. This problem with endogenous uncertainty requires the agent to find cost-minimizing paths to targets, which move to new locations when being reached. The costs of paths are influenced by prior paths.

**Experimental setup**   We compare SRL against two baselines: SIL, a structured imitation learning approach, and Proximal Policy Optimization (PPO), an unstructured RL algorithm. All algorithms use identical COAML-pipelines; SRL and PPO also share the same critic architectures to ensure a fair comparison. We select SIL due to its methodological proximity to SRL and its strong performance in combinatorial settings [Baty et al., 2024, Jungel et al., 2024]. We include PPO for its stability and compatibility with our pipeline architecture [Schulman et al., 2017]. Alternatives such as Soft Actor-Critic would require substantial modifications to the actor and critic architectures, as would

neural CO-methods [e.g., Bello et al., 2017] and Q-value-based optimization [e.g., Xu et al., 2025]. To keep this analysis concise, we concentrate on comparing COAML-pipelines using different learning paradigms and leave an in-depth comparison of architectures to future work. As performance references, we include two additional baselines: an expert policy and a greedy policy. In the static DVSP, the expert has access to the complete problem instance and thus represents the offline optimum. In contrast, in the dynamic DAP and GSPP, traceability constraints limit information access, and the expert corresponds to the best possible online policy, given the sequential nature of the decision process. We train all algorithms using the same number of episodes, employing environment-specific train/validation/test splits. We tune hyperparameters per algorithm and environment, using the PPO-optimized episode numbers consistently across methods. Each algorithm is retrained using ten random seeds. Appendix C provides further details on the experimental setup and baselines.

**Numerical results**   We present results for the dynamic environments in Figure 4 and Figure 5. We display the performance of final models on the train and test-datasets to measure algorithmic performance, and the development of validation rewards during training to highlight convergence behavior. While SRL performs comparably to SIL in the DVSP, it outperforms SIL in the DAP by 8% and in the GSPP by 78%, even surpassing the online optimum by 79% in the latter. In the DAP, the online optimum remains above the performance of SRL and SIL. These results highlight two key limitations of imitation learning: i) its performance is bounded by that of the expert policy; and ii) it lacks exploration to learn policies that enable escape from suboptimal states. In contrast, PPO consistently underperforms across all environments, struggling to reach greedy policies – SRL outperforms it by 16% in the DVSP, 77% in the DAP, and 92% in the GSPP. This poor performance highlights the challenges faced by unstructured RL in combinatorial action spaces. Overall, these performance gains show the superiority of SRL over SIL and PPO.

We observe notable differences in convergence speed. PPO converges slowest, requiring approximately 200 episodes in the DVSP and 160 in the GSPP to reach a performance plateau. In contrast, SRL and SIL converge at similar rates in the DVSP, while SIL converges approximately 150 episodes earlier in the DAP and 10 episodes earlier in the GSPP. This delay for SRL is expected, as it learns purely from interaction, without access to expert demonstrations.

Stability metrics in Table 1 further explain these trends. PPO shows the highest variance across all environments – up to $80\times$ higher than SRL and $40\times$ higher than SIL – reflecting known limitations of unstructured RL in combinatorial settings [e.g., Enders et al., 2023, Hoppe et al., 2024]. In contrast, SRL and SIL exhibit consistently low variance, underscoring the robustness of structured approaches.

**Discussion**   This robustness comes at a computational cost. SRL requires about 30 minutes of training per environment, compared to shorter runtimes for SIL and PPO. The difference arises from CO-layer usage: PPO invokes it only twice per update, versus $20\times$ in SIL and up to $61\times$ in SRL. Runtime also varies with CO-layer complexity – e.g., GSPP has a simple layer, leading to similar runtimes, while the more complex layer in DVSP increases SRLs runtime. The DAP runtime is further impacted by an expensive simulation and the use of two Q-networks. Although early stopping could mitigate this, the findings highlight a core limitation: SRLs computational cost scales with CO-layer complexity.

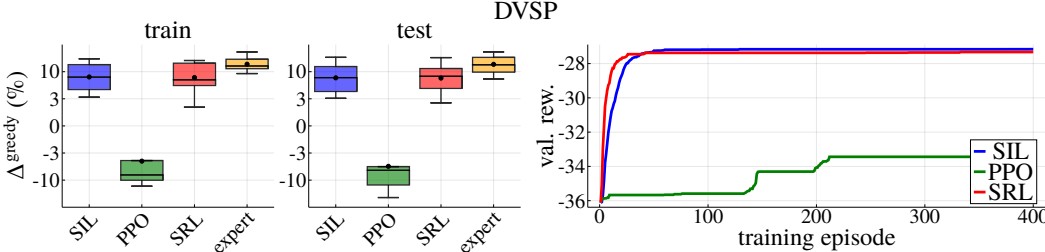

Figure 4: DVSP results. Left: final train and test-performance compared to greedy ($\Delta^{\text{greedy}}$); right: validation performance during training; averaged over 10 random model initializations.

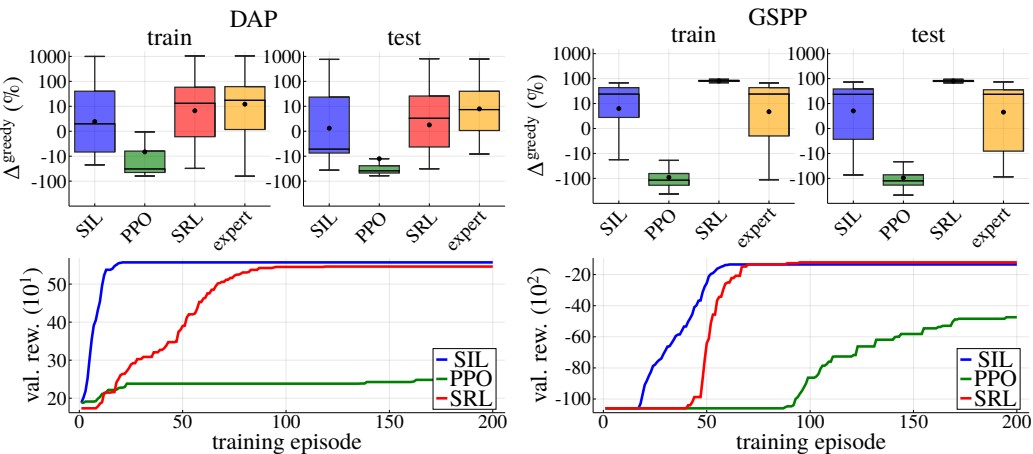

Figure 5: DAP and GSPP results. Left: final train and test-performance compared to greedy ($\Delta^{\text{greedy}}$); right: validation performance during training; averaged over 10 random model initializations.

Table 1: Standard deviation of validation rewards during training, final testing rewards over 10 random model initializations, and training time of algorithms in the DVSP, DAP and GSPP.

| Algorithm | DVSP | | | DAP | | | GSPP | | |
|---|---|---|---|---|---|---|---|---|---|
| | train | test | time | train | test | time | train | test | time |
| SIL | 0.3 | 0.4 | 12m | 0.8 | 11.9 | 3m | 39.3 | 1.1 | 11m |
| PPO | 5.8 | 5.6 | 3m | 5.4 | 13.5 | 5m | 105.8 | 47.0 | 10m |
| SRL | 0.3 | 0.3 | 31m | 1.8 | 1.9 | 31m | 72.1 | 0.6 | 34m |

Overall, the observations for our dynamic experiments align with our static experiments (Appendix E). SRL consistently matches SIL in performance and convergence, while PPO underperforms across all metrics. Runtimes follow the same pattern, increasing with CO-layer complexity.

In summary, our results yield three takeaways for real-world deployments: i) unstructured RL lacks the stability required for practical use. ii) SIL is limited to settings with simple dynamics and access to expert demonstrations. iii) Given the expected model and layer complexity in real-world settings, SRL offers a scalable and effective alternative solution approach at the price of a (reasonable) computational overhead when comparing it to SIL.

## 4 Conclusion

In this paper, we address combinatorial MDPs (C-MDPs), which present substantial challenges to current RL algorithms, despite being common in many industrial applications. Utilizing the framework of COAML-pipelines, we propose *Structured Reinforcement Learning* (SRL), a primal-dual algorithm using Fenchel-Young losses to train COAML-pipelines in an end-to-end fashion, thereby learning policies for C-MDPs using collected experience only. We compare SRL to Structured Imitation Learning (SIL) and unstructured RL in three static and three dynamic environments, representing typical industrial problem settings with combinatorial action spaces. The performance of SRL is competitive to SIL in the static environments and up to 78% better in the dynamic environments. SRL consistently outperforms unstructured RL by up to 92%, additionally being more stable and converging quicker, at the cost of higher computational effort.

## Acknowledgments and Disclosure of Funding

We thank the BAIS research group at TUM for valuable comments and discussions. The work of Heiko Hoppe was supported by the Munich Data Science Institute with a Linde/MDSI PhD Fellowship.

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

# A    Combinatorial Optimization-augmented Machine Learning pipelines

In a C-MDP, we are usually confronted with a high-dimensional state space $\mathcal{S}$ and a high-dimensional and combinatorial action space $\mathcal{A}(s)$. Processing the former requires an architecture capable of generalizing well across states and inferring information from contextual information. Neural networks, or more generally statistical models, are known to have these properties, which rule-based decision systems or combinatorial optimization methods commonly lack. In contrast, addressing combinatorial action spaces is challenging for statistical models: Using traditional approaches, every feasible action would have to correspond to one output node of the neural network, which then estimates a probability of choosing that action given the state. Such a network design is already challenging due to the state-dependent action space $\mathcal{A}(s)$, and furthermore impractical due to the large dimensionality of $\mathcal{A}(s)$. While the use of problem-specific neural networks of multi-agent approaches is possible, it is far easier to employ a combinatorial optimization to select a feasible action. Optimization methods have three advantages for this setting: i) they ensure feasibility of the selected action; ii) they scale to high-dimensional action spaces far better than plain neural network architectures; and iii) they naturally explore the action space by searching for an optimal solution iteratively.

Since we have established methods for handling both high-dimensional state spaces and high-dimensional, combinatorial action spaces, we can integrate these methods to leverage their combined strengths. This is the core idea behind CO-augmented Machine Learning-pipelines. In such pipelines, a statistical model $\varphi_w$, typically implemented as a neural network with parameters $w$, encodes the state $s$ to estimate a score vector $\theta = \varphi_w(s)$. A combinatorial optimization solver $f$ then uses these scores as coefficients in a linear objective function to compute an action by solving the following combinatorial optimization:

$$f(\theta, s) : a = \operatorname*{argmax}_{\tilde{a} \in \mathcal{A}(s)} : \langle \theta, \tilde{a} \rangle.$$

Through this integration, $f$ effectively becomes part of the actor model, commonly referred to as the CO-layer $f$. Importantly, the score vector $\theta$ is latent, i.e., it is not observed directly, nor are its true values typically known. When training a COAML-pipeline in an end-to-end fashion, we rely solely on observed outcomes, without explicit supervision on $\theta$. The resulting policy of such a pipeline can be formalized as $\pi_w(\cdot|s) := \delta_{f(\varphi_w(s),s)}$, representing a Dirac distribution centered on the action output by $f$.

As a motivating example, consider the Warcraft Shortest Paths Problem [Vlastelica et al., 2020], illustrated in Figure 6 and detailed in Appendix D. In this setting, the state $s$ is given by a map image with dimensions $96 \times 96 \times 3$ pixels. The task is to find the cost-minimizing path from the top-left to the bottom-right corner of the map, which corresponds to the action $a$. The path cost is determined by the terrain the agent traverses, with terrain types encoded by specific color codes. To solve this task, the state must first be encoded into a structured representation. We employ a convolutional neural network to estimate scores for each cell in a $12 \times 12$ grid. These scores $\theta$ are then used by Dijkstras algorithm to compute the path that minimizes the cumulative cell costs. Notably, neither the convolutional network nor Dijkstras algorithm alone can solve the raw WSPP map image. However, the COAML-pipeline efficiently learns to find cost-minimal paths through end-to-end training, combining the strengths of both components.

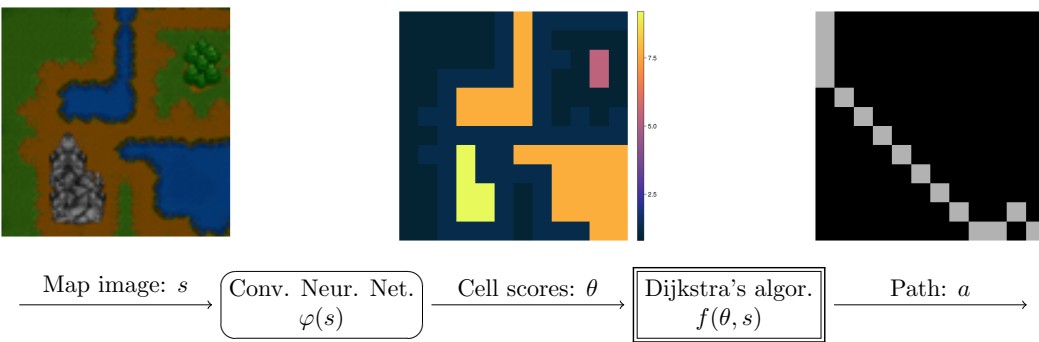

Figure 6: COAML-pipeline for the Warcraft Shortest Paths Problem: a convolutional neural network estimates cell scores based on image pixels. The CO-layer applies Dijkstra's algorithm on the scores to create a path between the top left and the bottom right corner.

# B Proofs

In Algorithm 1, we use two different regularization functions from the literature on the distribution simplex $\Delta^{\mathcal{A}}$. We start by introducing these regularizations. The literature considers (sub)gradients of these functions only in the (relative) interior of the probability simplex. In order to be able to work with sampled distributions, we extend them to the boundary of the simplex. We can then prove Proposition 2.

## B.1 Two regularizations: negentropy and sparse perturbation.

The first regularization is the *negentropy* [Blondel et al., 2020]:

$$\Omega_\Delta(q) = \sum_{a \in \mathcal{A}} q_a \log(q_a) + \mathbb{I}_{\Delta^{\mathcal{A}}}(q),$$

where $\mathbb{I}_{\Delta^{\mathcal{A}}}$ is the characteristic function of the set $\Delta^{\mathcal{A}}$, leading to the distribution

$$\nabla \Omega_\Delta^*(\gamma) = \left( e^{\gamma_a - A_\Delta(\gamma)} \right)_{a \in \mathcal{A}} \quad \text{where} \quad A_\Delta(\gamma) = \log \left( \sum_{a' \in \mathcal{A}} \exp(\gamma_{a'}) \right). \tag{9}$$

The second is the Fenchel conjugate of the *sparse perturbation* $\Omega_{\varepsilon,\Delta} := F_{\varepsilon,\Delta}^*$ [Bouvier et al., 2025], with

$$F_{\varepsilon,\Delta}(\gamma) = \mathbb{E}_Z[\max_{a \in \mathcal{A}} \gamma_a + \varepsilon Z^\top a] = \mathbb{E}_Z[\max_{q \in \Delta^{\mathcal{A}}} (\gamma + \varepsilon A^\top Z)^\top q],$$

where $Z \in \mathbb{R}^d$ is a random variable, typically a standard Gaussian. The resulting distribution is

$$\nabla \Omega_{\varepsilon,\Delta}^*(\gamma) = \nabla F_{\varepsilon,\Delta}(\gamma) = \mathbb{E}_Z[\underset{q \in \Delta^{\mathcal{A}}}{\operatorname{argmax}} (\gamma + \varepsilon A^\top Z)^\top q]. \tag{10}$$

Recall that the negentropy can be expressed as a variant of the conjugate of the sparse perturbation, when we take $Z$ distributed according to a Gumbel law.

## B.2 Extension on the boundary

The expectation in the right-hand side of Equation (10) naturally extends $\nabla \Omega_{\varepsilon,\Delta}^*$ in $\left( \mathbb{R} \cup \{-\infty\} \right)^{\mathcal{A}}$. Indeed, for any $\gamma \in \left( \mathbb{R} \cup \{-\infty\} \right)^{\mathcal{A}}$, and irrespective of the perturbation $Z \in \mathbb{R}^d$, any action $a$ satisfying $\gamma_a = -\infty$ will never appear in the argmax. This observation allows us to formally define the effective support of $\gamma$ as $\hat{\mathcal{A}}(\gamma) = \{a \in \mathcal{A} \mid \gamma_a > -\infty\}$, we get

$$\nabla \Omega_\Delta^*(\gamma) = \begin{cases} 0, & \text{for } a \in \mathcal{A} \setminus \hat{\mathcal{A}}(\gamma), \\ \nabla \Omega_{\Delta^{\hat{\mathcal{A}}(\gamma)}}^*(\hat{\gamma})_a, & \text{for } a \in \hat{\mathcal{A}}(\gamma), \end{cases} \tag{11}$$

where $\hat{\gamma} \in \mathbb{R}^{\hat{\mathcal{A}}(\gamma)}$ is the vector of finite components of $\gamma$, and $\nabla \Omega_{\Delta^{\hat{\mathcal{A}}(\gamma)}}^*(\hat{\gamma})_a$ is the component indexed by $a$ of the vector $\nabla \Omega_{\Delta^{\hat{\mathcal{A}}(\gamma)}}^*(\hat{\gamma})$ which belongs to $\Delta^{\hat{\mathcal{A}}(\gamma)}$. In the other way round, we extend $\partial \Omega_\Delta$ at the relative boundary of the simplex $\Delta^{\mathcal{A}}$ as follows. Let $q \in \Delta^{\mathcal{A}}$ be a sparse distribution (with some null components). In a similar way, we introduce the set $\hat{\mathcal{A}}(q) = \{a \in \mathcal{A} \mid q_a > 0\}$, and define

$$\partial \Omega_\Delta(q) \ni \gamma = \begin{cases} -\infty & \text{for } a \in \mathcal{A} \setminus \hat{\mathcal{A}}(q) \\ \hat{\gamma}_a, \hat{\gamma} \in \partial \Omega_{\Delta^{\hat{\mathcal{A}}(q)}}(\hat{q}) & \text{for } a \in \hat{\mathcal{A}}(q), \end{cases} \tag{12}$$

where $\hat{q} \in \Delta^{\hat{\mathcal{A}}(q)}$ is the sparse distribution seen as a dense distribution in the distribution simplex corresponding to its support.

## B.3 Proof of Proposition 2

*Proof of Proposition 2.* We go through the steps of Equations (8), and show that we indeed recover the actor update of Algorithm 1.

For step (8a), let $\theta^{(t)}$ be a given vector in $\mathbb{R}^d$, the distribution we sample from involves the gradient of the conjugate of the sparse perturbation

$$\nabla\Omega_{\varepsilon,\Delta}^*(A^\top\theta^{(t)}) = \mathbb{E}_Z\big[\operatorname*{argmax}_{q\in\Delta^\mathcal{A}}\langle A^\top\theta^{(t)} + \varepsilon A^\top Z|q\rangle\big] = \mathbb{E}_Z\big[\operatorname*{argmax}_{q\in\Delta^\mathcal{A}}\langle\big((\theta^{(t)} + \varepsilon Z)^\top a\big)_{a\in\mathcal{A}}|q\rangle\big].$$

Note that the argmax in the right-hand side is almost surely a Dirac because $a\mapsto(\theta^{(t)} + \varepsilon Z)^\top a$ is almost surely injective. Since the argmax returns a Dirac almost surely, we can equivalently rewrite the expectation as a probability for $a\in\mathcal{A}$:

$$\nabla\Omega_{\varepsilon,\Delta}^*(A^\top\theta^{(t)})_a = \Big[\mathbb{E}_Z\big[\operatorname*{argmax}_{q\in\Delta^\mathcal{A}}\langle\big((\theta^{(t)} + \varepsilon Z)^\top a'\big)_{a'\in\mathcal{A}}|q\rangle\big]\Big]_a,$$

$$= \mathbb{P}_Z\big(\operatorname*{argmax}_{a'\in\mathcal{A}}(\theta^{(t)} + \varepsilon Z)^\top a' = a\big).$$

Therefore, using the computations above, the step (8a) of sampling $(a_i^{(t+\frac{1}{2})})_{i\in[m]}\sim_{\text{i.d.}}\nabla\Omega_{\varepsilon,\Delta}^*(A^\top\theta^{(t)})$ is equivalent to sampling $(Z_i)_{i\in[m]}$, and computing for each $a_i^{(t+\frac{1}{2})} = \operatorname{argmax}_{a\in\mathcal{A}}(\theta^{(t)} + \varepsilon Z_i)^\top a$. This is precisely the first step in the actor update in Algorithm 1.

Step (8b) is explicit. The (empirical) sparse distribution $\hat{q}_m^{(t+\frac{1}{2})} = \frac{1}{m}\sum_{i=1}^m\delta_{a_i^{(t+\frac{1}{2})}}$ lies on the (relative) boundary of the simplex, as it assigns zero probability to actions not selected among the samples. For each $a$ in $\mathcal{A}$, let $k_a$ denote the number of samples $i$ such that $a_i^{(t+\frac{1}{2})} = a$.

In step (8c), we use the negentropy as regularization function $\Omega_\Delta$ over the distribution simplex $\Delta^\mathcal{A}$. Using the extension defined above, $\gamma_m^{(t+\frac{1}{2})}\in\partial\Omega_\Delta(\hat{q}_m^{(t+\frac{1}{2})})$ is such that there exists $\alpha\in\mathbb{R}$, such that for $a\in\mathcal{A}$,

$$\big(\gamma_m^{(t+\frac{1}{2})} + \frac{1}{\tau}\gamma_\beta\big)_a = \alpha + \ln(k_a) + \frac{1}{\tau}Q_{\psi_\beta}(a)$$

where $\ln(0)$ is taken equal to $-\infty$.

In step (8d), using again the extension of $\nabla\Omega_\Delta^*$ in $(\mathbb{R}\cup\{-\infty\})^\mathcal{A}$, Equation (9), and defining the normalization constant $\mathcal{Z}_{m,\beta} := \sum_{a\in\mathcal{A}}\exp\big(\ln(k_a) + \frac{1}{\tau}Q_{\psi_\beta}(a)\big)$,

$$\mu^{(t+1)} = A\nabla\Omega_\Delta^*\big(\gamma_m^{(t+\frac{1}{2})} + \frac{1}{\tau}\gamma_\beta\big) = \sum_{a\in\mathcal{A}}a\frac{k_a e^{\frac{1}{\tau}Q_{\psi_\beta}(a)}}{\mathcal{Z}_{m,\beta}} = \operatorname*{softmax}_{i\in[m]}\left(\frac{1}{\tau}Q_{\psi_\beta}(a_i^{(t+\frac{1}{2})})\right).$$

We thus recover the target action $\hat{a}$ defined by Equation (4).

Last, step (8e) can be written using Fenchel duality results [Blondel et al., 2020]

$$\theta^{(t+1)}\in\partial\Omega_{\varepsilon,\mathcal{C}}(\mu^{(t+1)}) \iff \theta^{(t+1)}\in\operatorname*{argmin}_\theta\mathcal{L}_{\Omega_{\varepsilon,\mathcal{C}}}(\theta;\mu^{(t+1)}),$$

where $\mathcal{L}_{\Omega_{\varepsilon,\mathcal{C}}}$ is the Fenchel-Young loss generated by $\Omega_{\varepsilon,\mathcal{C}}$. This is precisely the last step in the actor update in Algorithm 1, using the perturbed optimizer framework to define the regularization function, as detailed in Berthet et al. [2020].

This completes the proof that the primal-dual updates recover the actor update steps in Algorithm 1. $\square$

# C Experiments

In the following, we outline the setup and design of our experiments and explain the benchmark algorithms we used in all environments.

## C.1 Experimental setup

We conduct all experiments on the same hardware and use the same general method of conducting experiments across environments. We use the same results metrics for all algorithms and environments, ensuring comparability of the algorithms. In general, our experiments are reproducible using modest hardware equipment.

### C.1.1 Hardware setup

We conduct all experiments on a MacBook Air M3, using the Julia programming language. Given the usually small neural networks required for COAML-pipelines in our environments, the experiments take between 3 and 90 minutes. No external computing resources were required for running the experiments. This setup is the same as the one we used for all algorithmic development.

### C.1.2 Hyperparameters

We present an overview over the hyperparameters of the algorithms in Table 2. For the RL algorithms, an episode consists of testing the algorithm's performance, collecting experience in the environment, and performing a number of updates, specified as iterations. For SIL, episodes usually correspond to epochs, an epoch being a complete pass of the training dataset.

### C.1.3 Experiment conduction

We separate all instances into a train, validation, and test dataset. We create the training dataset for SIL by applying the expert policy to the training instances and storing the solutions. To tune the hyperparameters, we use the same random model initialization for SIL, PPO, and SRL. We tune the number of episodes and the number of iterations per episode using PPO, as it is typically the most constrained in terms of iterations and requires the largest number of episodes due to its on-policy, unstructured nature. This setup favors the baselines – particularly PPO – since SRL often converges more quickly but incurs higher computational cost per episode. As a result, PPO holds a natural advantage in runtime comparisons.

We then use the same number of episodes and iterations to train both PPO and SRL, and adjust the number of epochs for SIL to ensure that all methods perform approximately the same number of update steps overall. In most cases, this results in an equal number of episodes and epochs. For the DAP and GSPP, however, we reduce the number of epochs to account for the large size of the training

Table 2: Overview over hyperparameters included in the algorithms. Not all hyperparameters are used in all environments.

| Hyperparameter | SIL | PPO | SRL |
|---:|:---:|:---:|:---:|
| Episode number | Yes | Yes | Yes |
| Iterations number | Yes | Yes | Yes |
| Batch size | Yes | Yes | Yes |
| Learning rate actor (incl. schedule) | Yes | Yes | Yes |
| Learning rate critic(s) (incl. schedule) | No | Yes | Yes |
| Episodes training critic only | No | Yes | Yes |
| Replay buffer size | No | Yes | Yes |
| Exploration standard dev. $\sigma_f$ (incl. schedule) | No | Yes | Yes |
| No. samples for $\widehat{a}$ | No | No | Yes |
| Standard dev. $\sigma_b$ for $\widehat{a}$ (incl. schedule) | No | No | Yes |
| Temperature param. $\tau$ (incl. schedule) | No | No | Yes |
| No. samples for $\mathcal{L}_\Omega(\theta; \widehat{a})$ | Yes | No | Yes |
| Standard dev. $\varepsilon$ for $\mathcal{L}_\Omega(\theta; \widehat{a})$ | Yes | No | Yes |

dataset. We tune the exploration standard deviation $\sigma_f$, the perturbation standard deviation $\sigma_b$, the temperature parameter $\tau$, and the learning rate – typically shared between actor and critic, or set slightly higher for the critic – using a grid search for each algorithm. Each grid search involves between 3 and 30 training runs. All other hyperparameters do not require detailed tuning.

Once the optimal hyperparameters are identified, we run each algorithm with ten randomly initialized actor (and, where applicable, critic) models. After each episode or epoch, the actor is evaluated on the training and validation datasets – or a subset thereof to improve efficiency. We save the actor model whenever it achieves the best performance observed so far. At the end of training, the best-performing actor model is restored and used for final evaluation on the training and test datasets.

### C.1.4 Results metrics

To compare performance, we run the best saved model of each algorithm after each run with different random model initializations on the train and the test dataset. We calculate the mean over the ten models per instance of the train and test dataset, using these mean per-instance rewards in the results boxplots. To compare convergence speed, we store the validation rewards over the course of training and calculate the mean across the ten runs per algorithm and environment per episode. In the lineplots, we display the highest mean validation reward achieved by the model so far for each training episode. We report the mean of the standard deviations across the validation and final test rewards of the ten runs in the tables. We further measure the time to run an algorithm in minutes, reporting that number in the tables as well. Finally, we calculate the overall mean reward of the final tests on the train and test dataset per algorithm and environment and display it in Appendix E.

### C.2 Algorithm specification

For training COAML-pipelines, we compare SRL to two benchmark algorithms: SIL and PPO. SIL uses Fenchel-Young losses like SRL, but relies on expert imitation instead of reinforcement learning. Due to its methodological proximity and empirical performance [e.g., Baty et al., 2024, Jungel et al., 2024], it is a natural benchmark for SRL. PPO is an unstructured RL algorithm, which is well-known for its stability and performance. Therefore, it is the most sensible RL-benchmark for SRL. We also show why PPO is better suited than Soft Actor-Critic for training the COAML-pipelines used in our experiments.

### C.2.1 Structured Imitation Learning

Structured Imitation Learning is an imitation learning algorithm successfully applied in recent works such as Baty et al. [2024] and Jungel et al. [2024]. As an imitation learning approach, SIL requires a pre-collected training dataset consisting of states $s$ and corresponding expert actions $\bar{a}$. Since the algorithm has direct access to these expert actions, it does not rely on a critic to generate learning targets.

Training the actor model using SIL proceeds by iterating over the training dataset and updating the model using the Fenchel-Young loss $\mathcal{L}_\Omega(\theta; \bar{a})$, which compares the models unperturbed score vector $\theta$ to the expert action $\bar{a}$. This update step is structurally identical to that used in SRL, and we apply the same hyperparameters for the Fenchel-Young loss in both algorithms. The critical distinction is the source of the target action: while SRL derives its target action $\hat{a}$ from a critic, SIL directly uses the expert action $\bar{a}$ from the provided training dataset. In practice, SIL can be trained using mini-batches, though it often suffices to perform updates with individual state-action pairs.

Unlike online reinforcement learning methods, SIL does not interact with the environment during training and, accordingly, does not require an exploration standard deviation $\sigma_f$. This offline setup enhances sample efficiency and improves training stability. However, it introduces a limitation in multi-stage environments: since SIL exclusively observes expert trajectories, it cannot learn effective policies for situations outside the demonstrated paths. Consequently, if the agent deviates from the expert path during deployment, it may struggle to recover, potentially leading to sub-optimal decisions.

### C.2.2 Proximal Policy Optimization

PPO is a classical RL algorithm proposed by Schulman et al. [2017], to whom we refer for details. In the context of COAML-pipelines, PPO selects actions by perturbing the score vector $\theta$ using a Gaussian distribution $Z \sim N(\theta_j, \sigma_f)$, sampling a single perturbed score vector $\eta$, and calculating $a = f(\varphi_w(s), s)$. In its training, PPO considers the CO-layer $f$ to be part of the environment and treats the perturbed score vector $\eta$ as its action. It then calculates the loss function

$$\mathcal{L}(\varphi_w) = \mathbb{E}_{j \sim D} \left[ \min \left( \frac{\pi_{\varphi_w}(\eta_j|s_j)}{\pi_{\varphi_{\bar{w}}}(\eta_j|s_j)} \cdot A(s_j, \eta_j), \, \text{clip}\left( \frac{\pi_{\varphi_w}(\eta_j|s_j)}{\pi_{\varphi_{\bar{w}}}(\eta_j|s_j)}, 1 - \epsilon, 1 + \epsilon \right) \cdot A(s_j, \eta_j) \right) \right]$$

for transitions $j$ in replay buffer $D$.

Given the use of $\eta$ as an action, PPO considers the policy $\pi_{\varphi_w}(\eta|s)$, which is the probability of observing vector $\eta$ given state $s$ under the Gaussian distribution $Z \sim N(\theta, \sigma_f)$. In practise, the probability density function of $\eta$ given $Z$ is used. If we update using batches that contain transitions collected with different values for $\sigma_f$, we average $\sigma_f$ to improve stability. A key element of PPO is the policy ratio, which should ensure proximity between new and old policies via clipping

$$\frac{\pi^{\varphi_w}(\eta|s)}{\pi^{\varphi_{\bar{w}}}(\eta|s)}.$$

The policy ratio is the probability of observing $\eta$ given the current actor $\varphi_w$ divided by the probability of observing $\eta$ given the old actor $\varphi_{\bar{w}}$. The old actor $\varphi_{\bar{w}}$ is the network used to collect the experience considered in the current update.

PPO clips the policy ratio using the clipping ratio $\epsilon$ to ensure that the new policy does not deviate into untrusted regions far away from the old policy. Constrained by the clipping, the target of a PPO update is the maximization of the advantage $A(s, \eta) = Q(s, \eta) - V(s)$. Since $V(s) = Q(s, \theta)$, the advantage is the difference in value gained by executing the action corresponding to $\eta$ instead of the action corresponding to $\theta$ given the old policy $\pi^{\varphi_{\bar{w}}}$. Finally, PPO calculates and applies the gradients $\nabla_w \varphi_w(s)$ to $\varphi_w$.

For estimating the Q-values and V-values, we we use the same critic architectures as for SRL. Despite being an on-policy algorithm, PPO can use a replay buffer, although that is usually smaller than for off-policy algorithms.

### C.2.3 Benchmark reasoning

We choose PPO over Soft Actor-Critic (SAC) [Haarnoja et al., 2018, 2019] for the following reasons:

**Actor Network Structure** SAC requires the actor to output both the mean and the standard deviation of a continuous action distribution in order to perform entropy regularization. This would necessitate neural networks with two outputs—$\theta$ and $\sigma_f$. In contrast, PPO allows us to manually set $\sigma_f$, avoiding the need for a separate network head or specialized actor architecture. This ensures consistency across algorithms and avoids introducing additional sources of divergence.

**Critic Differentiability Constraints** SAC requires the critic to be differentiable with respect to the actor. However, our critic takes the form $Q_{\psi_\beta}(s, a)$ and operates directly on actions $a$, making such differentiation infeasible. Adapting the critic to work with $\theta$ or to be decomposable would require fundamentally different architectures, which, in early experiments, led to significantly worse performance. Moreover, directly differentiating through $Q_{\psi_\beta}(s, a)$ would require specialized structured loss functions, which are not yet available and remain an open problem for future work.

**Entropy Regularization in Combinatorial Action Spaces** SAC inherently relies on entropy regularization of the action distribution. In our setting, this would regularize the distribution of $\theta$, while a regularization over the combinatorial actions $a$ is actually needed, which is not directly feasible given our pipeline. Relying on entropy to adjust $\sigma_f$ in this setup could lead to instability or poor local optima. We thus avoid this risk by setting $\sigma_f$ manually and leave the development of suitable entropy regularization schemes for combinatorial action spaces to future work.

# D   Environment specification

In the following, we provide a description of the six environments we use to test SRL. For each environment, we explain the environment specification, the design of the expert and the greedy policies, how the COAML-pipeline is specified, how the critic is specified, and what hyperparameters we use for each algorithm in the environment. We will start with the static environments, followed by the dynamic environments.

## D.1   Warcraft Shortest Paths Problem

The *Warcraft Shortest Path Problem* is a popular benchmark in the literature on COAML-pipelines, introduced by Vlastelica et al. [2020].

**Environment specification**   The goal is to find the shortest path between the top left and the bottom right corners of a map. The observed state $s$ is a map image of $96 \times 96 \times 3$ pixels representing the map as a 3D array of pixels. Each map is decomposed into a $12 \times 12$ grid of cells, with each cell having a cost. The cost depends on the difficulty of the associated terrain. Each terrain has a specific color, allowing for the inference from pixels to costs. The costs themselves are not observed in the state, but hidden from the agent. The action space is the set of all paths from the top left corner to the bottom right corner of the map. The reward is the negative total cost of the path, i.e. the sum of the hidden costs of all cells in the path.

**Expert policy**   Using full knowledge of cell costs, the optimal path is computed using Dijkstra's algorithm on the cell costs [Dijkstra, 1959].

**Greedy policy**   The greedy policy is a straight path from the top left to the bottom right corner of the map, disregarding all cell costs.

**COAML-pipeline**   We use a similar pipeline as in Dalle et al. [2022]: the actor model $\varphi_w$ is a convolutional neural network, based on the logic of a truncated ResNet18, with output dimension $12 \times 12$ The CO-layer $f$ is Dijkstra's algorithm [Dijkstra, 1959], which computes the shortest path between the two corners of the map.

**Critic specification**   Since the problem is static and the rewards are deterministic given an action, we do not employ a critic neural network in the WSPP. We assume access to the black-box cost function and use this function as our critic.

**Hyperparameters**   We present the hyperparameters utilized for the WSPP in Table 3. The number of iterations correspondents to the size of the train dataset.

Table 3: Overview over hyperparameters in the WSPP.

| Hyperparameter | SIL | PPO | SRL |
|---|---|---|---|
| Episode number | 200 | 200 | 200 |
| Iterations number | 120 | 120 | 120 |
| Batch size | 60 | 20 | 60 |
| Learning rate actor (incl. schedule) | 1e-3 | 5e-4 $\rightarrow$ 1e-4 | 2e-3 $\rightarrow$ 1e-3 |
| Exploration standard dev. $\sigma_f$ (incl. schedule) | – | 0.1 $\rightarrow$ 0.05 | – |
| No. samples for $\widehat{a}$ | – | – | 40 |
| Standard dev. $\sigma_b$ for $\widehat{a}$ (incl. schedule) | – | – | 0.1 $\rightarrow$ 0.05 |
| Temperature param. $\tau$ (incl. schedule) | – | – | 0.1 $\rightarrow$ 0.01 |
| No. samples for $\mathcal{L}_\Omega(\theta; \widehat{a})$ | 20 | – | 20 |
| Standard dev. $\varepsilon$ for $\mathcal{L}_\Omega(\theta; \widehat{a})$ | 0.05 | – | 0.05 |

## D.2 Single Machine Scheduling Problem

The *Single Machine Scheduling Problem* that we consider is a static industrial problem setting with a large combinatorial action space, introduced by Parmentier and T'Kindt [2023].

**Environment specification**  An instance of the *single machine scheduling problem* requires scheduling a total of $n \in [50, 100]$ jobs on a single machine. Each job $j \in [n]$ has a given processing time $p_j$ and an release time $r_j$, prior to which job $j$ cannot be initiated. The machine is limited to processing exactly one job at any given moment. Once processing of a job begins, it must run to completion without interruption, as preemption is prohibited. The objective is the determination of an optimal scheduling sequence as a permutation $s = (j_1, ..., j_n)$ of the jobs in $[n]$ that minimizes the total completion time $\sum_j C_j(s)$, with $C_j(s)$ being the completion time of job $j$. Specifically, for the first job in the sequence, we have $C_{j_1}(s) = r_{j_1} + p_{j_1}$, while for subsequent jobs where $k > 1$, the completion time is calculated as $C_{j_k}(s) = \max\left(r_{j_k}, C_{j_{k-1}}(s)\right) + p_{j_k}$.

**Expert policy**  For instances with up to $n = 110$ jobs a branch-and-memorize algorithm [Shang et al., 2021] is used as an exact algorithm to compute optimal solutions.

**Greedy policy**  The greedy policy builds a greedy sequence by sorting jobs by increasing release times. Ties are broken by processing jobs with lower processing times first.

**COAML-pipeline**  The actor model $\varphi_w$ is a simple generalized linear model with input dimension 27 and output dimension 1. For each job $j \in [n]$ we compute the corresponding feature vector $x_j \in \mathbb{R}^{27}$. The features used in this model are taken from Parmentier and T'Kindt [2023]. This allows to compute $(\theta)_{j \in [n]} = (\varphi_w(x_j))_{j \in [n]}$ by applying the linear model in parallel to every job. The CO-layer $f$ is the ranking operator, which can be formulated as a linear optimization problem:

$$f: \theta \mapsto \mathrm{ranking}(\theta) = \operatorname*{argmax}_{y \in \sigma(n)} \theta^\top y,$$

where $\sigma(n)$ is the set of permutations of $[n]$.

**Critic specification**  In this static environment, we again do not employ critic neural networks, but assume having access to the black-box cost function. This assumption is realistic, since evaluating the duration of a schedule it is easier than finding a schedule.

**Hyperparameters**  We present the hyperparameters utilized for the Single Machine Scheduling Problem (SMSP) in Table 4. The number of iterations correspondents to the size of the train dataset.

## D.3 Stochastic Vehicle Scheduling Problem

The *Stochastic Vehicle Scheduling Problem* that we consider is a static, stochastic problem setting with a large combinatorial action space, introduced by Parmentier [2022].

Table 4: Overview over hyperparameters in the SMSP.

| Hyperparameter | SIL | PPO | SRL |
|---|---|---|---|
| Episode number | 2000 | 2000 | 2000 |
| Iterations number | 420 | 420 | 420 |
| Batch size | 1 | 20 | 20 |
| Learning rate actor (incl. schedule) | 1e-3 | 5e-4 | 2e-3 $\to$ 1e-3 |
| Exploration standard dev. $\sigma_f$ (incl. schedule) | – | 0.01 | – |
| No. samples for $\widehat{a}$ | – | – | 40 |
| Standard dev. $\sigma_b$ for $\widehat{a}$ (incl. schedule) | – | – | 2.0 |
| Temperature param. $\tau$ (incl. schedule) | – | – | 1e3 |
| No. samples for $\mathcal{L}_\Omega(\theta; \widehat{a})$ | 20 | – | 20 |
| Standard dev. $\varepsilon$ for $\mathcal{L}_\Omega(\theta; \widehat{a})$ | 1.0 | – | 1.0 |

**Environment specification** The Stochastic Vehicle Scheduling Problem focuses on optimizing vehicle routes across time-constrained tasks in environments with stochastic delay. Each task $v \in \bar{V}$ is characterized by its scheduled start time $t_v^b$ and end time $t_v^e$ (where $t_v^e > t_v^b$). Vehicles can only perform tasks sequentially, with a travel time $t_{(u,v)}^{tr}$ required between the completion of task $u$ and start of task $v$.

Tasks can only be sequentially assigned to the same vehicle when timing constraints are satisfied:

$$t_v^b \geq t_u^e + t_{(u,v)}^{tr}$$

The problem can be represented as a directed acyclic graph $D = (V, A)$, where $V = \bar{V} \cup \{o, d\}$ includes all tasks plus two dummy origin and destination nodes. Arcs exist between consecutive feasible tasks, with every task connected to both origin and destination.

A feasible decision for this problem is therefore a set of disjoint $s - t$ paths such that all tasks are covered.

What distinguishes the stochastic variant is the introduction of random delays that propagate through task sequences. The objective becomes minimizing the combined cost of vehicle routes and expected delay penalties. The cost of a vehicle is denoted by $c_{\text{vehicle}}$, and the cost of a unit of delay $c_{\text{delay}}$. We consider multiple scenarios $s \in S$, where each task $v$ experiences an intrinsic delay $\gamma_v^s$ in scenario $s$. The total delay $d_v^s$ of a task $v$ accounts for both intrinsic delays and propagated delays from preceding task $u$ on the route, calculated as:

$$d_v^s = \gamma_v^s + \max(d_u^s - \delta_{u,v}^s, 0)$$

Here, $\delta_{u,v}^s$ represents the time buffer between consecutive tasks $u$ and $v$.

We train and test using $|\bar{V}| = 25$ tasks.

For more details about this environment specifications, we refer to Dalle et al. [2022]

Table 5: Overview over hyperparameters in the SVSP.

| Hyperparameter | SIL | PPO | SRL |
|---|---|---|---|
| Episode number | 200 | 200 | 200 |
| Iterations number | 50 | 50 | 50 |
| Batch size | 1 | 4 | 4 |
| Learning rate actor (incl. schedule) | 1e-3 | 1e-2 | 1e-2 → 5e-3 |
| Exploration standard dev. $\sigma_f$ (incl. schedule) | – | 0.5 → 0.1 | – |
| No. samples for $\widehat{a}$ | – | – | 20 |
| Standard dev. $\sigma_b$ for $\widehat{a}$ (incl. schedule) | – | – | 0.1 → 0.01 |
| Temperature param. $\tau$ (incl. schedule) | – | – | 1e4 → 1e2 |
| No. samples for $\mathcal{L}_\Omega(\theta; \widehat{a})$ | 20 | – | 20 |
| Standard dev. $\varepsilon$ for $\mathcal{L}_\Omega(\theta; \widehat{a})$ | 1.0 | – | 1.0 |

**Expert policy**    An anticipative solution can be computed by solving the following quadratic mixed integer program:

$$\min_{d,y} c_{\text{delay}} \frac{1}{|S|} \sum_{s \in S} \sum_{v \in V \setminus \{o,d\}} d_v^s + c_{\text{vehicle}} \sum_{a \in \delta^+(o)} y_a \tag{13a}$$

$$\text{s.t.} \sum_{a \in \delta^-(v)} y_a = \sum_{a \in \delta^+(v)} y_a \qquad \forall v \in V \setminus \{o,d\} \tag{13b}$$

$$\sum_{a \in \delta^-(v)} y_a = 1 \qquad \forall v \in V \setminus \{o,d\} \tag{13c}$$

$$d_v^s \geq \gamma_v^s + \sum_{\substack{a \in \delta^-(v) \\ a=(u,v)}} (d_u^s - \delta_{u,v}^s) \, y_a \qquad \forall v \in V \setminus \{o,d\}, \forall s \in S \tag{13d}$$

$$d_v^s \geq \gamma_v^s \qquad \forall v \in V \setminus \{o,d\}, \forall s \in S \tag{13e}$$

$$y_a \in \{0,1\} \qquad \forall a \in A \tag{13f}$$

This assumes knowledge of delay scenarios in advance; therefore it cannot be used in practical deployment, but serves as a perfect-information bound.

**Greedy policy**    The greedy policy for this problem is solving the deterministic variant of the problem instead, which only minimizes vehicle costs without taking into account delays. In this case, the problem is easily solved using a flow-based linear program formulation.

**COAML-pipeline**    For each arc $a$ in the graph, we compute a feature vector of size 20, containing information about the arc and delay propagation distribution along it. The actor model $\varphi_w$ is a generalized linear model, that is applied in parallel to all arcs in the graph. It therefore has input dimension 20 and output dimension 1. The CO-layer $f$ is a linear programming solver, which computes the optimal flow-based solution for the problem, replacing determinsitic arc costs by estimated scores from the actor model.

**Critic specification**    In this static environment, we employ sample average approximation instead of critic neural networks to evaluate the costs of an action. For this evaluation, we randomly draw 10 (for 25 tasks) or 50 (for 100 tasks) scenarios per instance, corresponding to delay realizations. We apply the policy to every scenario and estimate the total delay of the scenario. We then calculate the costs as the average delay across all scenarios. Since evaluating an action is considerably easier than generating one, this is a reasonable cost evaluation method given contextual information and stochasticity.

**Hyperparameters**    We present the hyperparameters utilized for the Stochastic Vehicle Scheduling Problem (SVSP) in Table 5. The number of iterations correspondents to the size of the train dataset.

## D.4    Dynamic Vehicle Scheduling Problem

The *Dynamic Vehicle Scheduling Problem* that we consider is a simplified variant of the *Dynamic Vehicle Routing Problem with Time Windows* introduced in the EURO-NeurIPS challenge 2022 [Kool et al., 2022].

**Environment specification**    The Dynamic Vehicle Scheduling Problem requires deploying a fleet of vehicles to serve customers that arrive dynamically over a planning horizon. At each time stage, we observe all unserved customers currently in the system, denoted by $V_t$. We then must: i) determine which customers to dispatch vehicles to; and ii) build vehicle routes starting from the depot to serve them. Each customer has a specific location and time that must be strictly respected. Vehicle routes must adhere to time constraints, with waiting allowed at customer locations without additional cost. The objective is to minimize the total travel cost across all vehicles. A key operational constraint is that customers approaching their deadline are designated as "must-dispatch" requiring immediate service to ensure feasibility. We denote by $V_t^{\text{md}} \subset V$ the set of must-dispatch customer. This mechanism

ensures all customers receive service within their time windows by the end of the planning horizon. An episode has 8 time steps.

Similarly to the stochastic vehicle scheduling problem described above, a feasible decision at time step $t$ can be viewed as a set of disjoint paths in an associated acyclic graph $D = (V_t, A_t)$.

**Expert policy** We compute the anticipative policy that constructs globally optimal routes. Assuming knowledge of all future customer arrivals, we obtain this policy by solving the static vehicle scheduling problem, then decomposing the solution into time-step-specific dispatching decisions.

**Greedy policy** The greedy policy for this problem is to dispatch all customers as soon as they appear; and optimize routes at each time step by solving a static vehicle scheduling problem.

**COAML-pipeline** We use a similar pipeline as introduced by Baty et al. [2024]. The actor model $\varphi_w$ is a generalized linear model with input dimension 14 and output dimension 1. This model is applied in parallel to each customer $v \in V_t$, to predict a prize $\theta_v$ from its feature vector of size 14.

The CO-layer $f$ is a static vehicle scheduling MIP solver, with arc costs $\theta$ predicted by the actor model:

$$
f: \theta \longmapsto
\begin{cases}
\arg\min_y \sum_{a=(u,v)\in A_t} (\theta_v - d_a) y_a \\[2mm]
\text{s.t.} \sum_{a\in\delta^-(v)} y_a = \sum_{a\in\delta^+(v)} y_a, & \forall v \in V_t \\[2mm]
\quad \sum_{a\in\delta^-(v)} y_a \leq 1, & \forall v \in V_t \\[2mm]
\quad \sum_{a\in\delta^-(v)} y_a = 1, & \forall v \in V_t^{\mathrm{md}} \\[2mm]
\quad y_a \in \{0,1\}, & \forall a \in A_t
\end{cases}
\tag{14}
$$

**Critic specification** In the DVSP, we use a single graph neural network as the critic. The network takes the solution graph as input and outputs a single value as the Q-value, using several graph convolutional layers, a global additive pooling layer, and finally several fully connected feedforward layers as its architecture. The solution graph is a a graphical representation of an action in the DVSP, whith all requests being nodes and vehicle routes being edges. For each node, we pass a feature vector into the critic network. These features are the same as for the actor, plus an indicator whether a request is postponable or not. For each edge, we pass the distance into the critic. We train the critic using ordinary Huber losses between $Q_{\psi_{\beta,k}}(s_t, a_t)$ and $y_t = r_t + \gamma \, Q_{\psi_{\beta,k}}(s_{t+1}, \pi(a_{t+1}))$. For this, we use the same transitions as immediately afterwards for the policy update, which we sample from the replay buffer.

Table 6: Overview over hyperparameters in the DVSP.

| Hyperparameter | SIL | PPO | SRL |
|---|---|---|---|
| Episode number | 400 | 400 | 400 |
| Iterations number | 100 | 100 | 100 |
| Batch size | 1 | 1 | 4 |
| Learning rate actor (incl. schedule) | 1e-3 | 1e-3 → 5e-4 | 1e-3 → 2e-4 |
| Learning rate critic(s) (incl. schedule) | – | 1e-2 → 5e-4 | 2e-3 → 2e-4 |
| Replay buffer size | – | 12000 (2000 eps.) | 120000 (20000 eps.) |
| Exploration standard dev. $\sigma_f$ (incl. schedule) | – | 0.5 → 0.05 | 0.1 |
| No. samples for $\widehat{a}$ | – | – | 40 |
| Standard dev. $\sigma_b$ for $\widehat{a}$ (incl. schedule) | – | – | 1.0 → 0.1 |
| Temperature param. $\tau$ (incl. schedule) | – | – | 10 |
| No. samples for $\mathcal{L}_\Omega(\theta; \widehat{a})$ | 20 | – | 20 |
| Standard dev. $\varepsilon$ for $\mathcal{L}_\Omega(\theta; \widehat{a})$ | 0.01 | – | 0.01 |

**Hyperparameters**  We present the hyperparameters utilized for the DVSP in Table 6

## D.5  Dynamic Assortment Problem

The *Dynamic Assortment Problem* that we consider is a multi-stage problem with endogenous uncertainty and a large combinatorial action space. We use a version adapted from the Dynamic Assortment Optimization problem introduced by Chen et al. [2020]. The DAP is also related to recommender systems, as used by Dulac-Arnold et al. [2016].

**Environment specification**  We have $n = 20$ items $i \in I$, of which we can show an assortment $S$ of size $K = 4$ to a customer each time step. Each item has 4 features and a price, creating the feature vector $v$. The uniform customer calculates a score $\Theta$ per item using a hidden linear customer model $\Phi$ as $\Theta = v^\top \Phi$. The customer then estimates purchase probabilities for each item $i$ using the multinomial logit model

$$P(i|S) = \frac{\exp \Theta_i}{1 + \sum_{j \in S} \exp \Theta_j},$$

which includes the option of not buying an item. By sampling from the purchase probabilities, the customer purchases an item or no items. We receive the item's price as a reward $r(i)$. After purchasing an item, the third feature of that item is increased by a "hype" factor, which is decreased again over the subsequent 4 time steps. The fourth feature is increased by a "satisfaction" factor as a once-of increment. The other features and the price remain static, having been randomly initialized at the beginning of an episode. The hidden customer model $\Phi$ remains static globally. An episode has 80 time steps.

**Expert policy**  Due to the endogenous feature updates, finding a globally optimal policy is computationally intraceable. We therefore resort an online optimal policy as an approximation. Given knowledge of $v$ and $\Phi$, we enumerate all possible assortments $S$ of size $K$, calculating $P(i|S) \forall i \in S$ and then calculating the expected revenue of $S$ as $R(S) = \sum_{i \in S} r(i) \cdot P(i|S)$. We finally select the assortment $S$ with the highest expected revenue $R(S)$.

**Greedy policy**  The greedy policy sorts items $i$ by their price $r(i)$ and selects the $K$ items with the highest $r(i)$ as $S$.

**COAML-pipeline**  The actor model $\varphi_w$ is a 2-layer fully connected feedforward neural network with input dimension 10, hidden dimension 5, and output dimension 1. In addition to the feature vector $v$, its input is the current time step relative to the maximum episode length, the one-step change of the endogenous features, and the change of these features since the start of the episode. We perform one pass through $\varphi_w$ per $i$, generating a vector of scores $\theta$. The CO-layer $f$ ranks $\theta$ and selects the $K$ items corresponding to the highest $\theta$ as $S$.

Table 7: Overview over hyperparameters in the DAP.

| Hyperparameter | SIL | PPO | SRL |
|---|---|---|---|
| Episode number | 200 | 200 | 200 |
| Iterations number | 100 | 100 | 100 |
| Batch size | 1 | 4 | 4 |
| Learning rate actor (incl. schedule) | 1e-4 | 5e-3 | 1e-3 $\rightarrow$ 5e-4 |
| Learning rate critic(s) (incl. schedule) | – | 5e-3 | 1e-3 $\rightarrow$ 5e-4 |
| Replay buffer size | – | 1600 (20 eps.) | 8000 (100 eps.) |
| Exploration standard dev. $\sigma_f$ (incl. schedule) | – | 0.1 $\rightarrow$ 0.05 | 2.0 $\rightarrow$ 1.0 |
| No. samples for $\widehat{a}$ | – | – | 40 |
| Standard dev. $\sigma_b$ for $\widehat{a}$ (incl. schedule) | – | – | 2.0 $\rightarrow$ 1.0 |
| Temperature param. $\tau$ (incl. schedule) | – | – | 1.0 |
| No. samples for $\mathcal{L}_\Omega(\theta; \widehat{a})$ | 20 | – | 20 |
| Standard dev. $\varepsilon$ for $\mathcal{L}_\Omega(\theta; \widehat{a})$ | 1.0 | – | 1.0 |

**Critic specification**   Given the highly stochastic nature of this environment, we employ two critics: the first critic should approximate $R(S)$; it first processes $v$ and the relative time step, which is part of the vector for computational simplicity, using a feedforward layer with an output size 3 per $i \in S$. It then concatenates these intermediate outputs in a vector and feeds them through another feedforward layer with output size 1. It is trained by minimizing the Huber loss between the actual reward $r(i)$ of purchased item $i$ and its output. The second critic should approximate $Q^\pi(s_t, a_t) - r_t = \gamma \, Q^\pi(s_{t+1}, \pi(s_{t+1}))$. It receives all features that $\varphi_w$ receives plus a binary indicator whether $i \in S$ as input for all $i$, using a feedforward layer to estimate 5 hidden scores per $i$. After concatenating these scores, it estimates Q-values using a 2-layer feedforward neural network with hidden dimension 10 and output dimension 1. It is trained by minimizing the Huber loss between the on-policy returns $ret_t = \sum_{k=t+1}^{T} \gamma^{k-t} r_k$ and its output. Due to its on-policy nature, it receives shuffled transitions from the last episode instead of sampled transition from the replay buffer. After failing to converge in initial tests using both critics, we only employ the first critic in PPO. Given the good performance of the myopically optimal policy, and comparably good results when training SRL using only the first or both critics, this should not impede the performance PPO can reach substantially. Were it to converge properly, it should still display a performance similar to that of SIL.

**Hyperparameters**   We present the hyperparameters utilized for the DAP in Table 7

### D.6   Gridworld Shortest Paths Problem

The *Gridworld Shortest Paths Problem* that we consider is a dynamic problem with endogenous uncertainty and a large combinatorial action space. It is related to gridworld problems that are commonly used to investigate the scalability of RL algorithms [e.g., Chandak et al., 2019]. It is furthermore related to robot control tasks in discrete environments [e.g., Zhang et al., 2020].

**Environment specification**   We control a robot in a gridworld of size $20 \times 20$ with cells $(i, j) \in (I, J)$. Each time step, we need to find the best path between the current position of the robot and a target, the robot can move to all 8 neighbors of a cell if they exist. Upon reaching the target following the path, the target moves to a random new location and we transition to the next state. Each cell $(i, j)$ in the gridworld has six features, $v_{i,j}$ which are randomly initialized at the beginning of an episode and which remain constant for the remainder of the episode. The first three features $v_{i,j}^c$ determine the immediate costs of the cell $c_{i,j}$ via a fixed linear model $\Phi^c$ as $c_{i,j} = (v_{i,j}^c)^T \Phi^c$. The final three features $v_{i,j}^\rho$ determine the change of a cost parameter $\rho$ via another fixed linear model $\Phi^\rho$ as $\Delta\rho_{i,j} = (v_{i,j}^\rho)^T \Phi^\rho$. The cost of a path $C$ is calculated as the sum of all $c_{i,j}$ of traversed cells times $\rho$. The cost parameter $\rho$ is subsequently updated by multiplying itself with one plus the sum of all $\Delta\rho_{i,j}$ of traversed cells. The presence of $\rho$ introduces strong endogeneity to the problem: a path minimizing the sum of $c_{i,j}$ does not have to be globally optimal, but minimizing the immediate costs has to be balanced with minimizing $\rho$. The linear models $\Phi^c$ and $\Phi^\rho$ remain hidden for agents,

Table 8: Overview over hyperparameters in the GSPP.

| Hyperparameter | SIL | PPO | SRL |
|---|---|---|---|
| Episode number | 200 | 200 | 200 |
| Iterations number | 100 | 100 | 100 |
| Batch size | 1 | 1 | 4 |
| Learning rate actor (incl. schedule) | 1e-4 | 5e-4 | 1e-3 $\rightarrow$ 5e-4 |
| Learning rate critic(s) (incl. schedule) | – | 5e-4 | 1e-3 $\rightarrow$ 5e-4 |
| Episodes training critic only | – | 40 | 40 |
| Replay buffer size | – | 2000 (20 eps.) | 10000 (100 eps.) |
| Exploration standard dev. $\sigma_f$ (incl. schedule) | – | 0.05 | 0.05 |
| No. samples for $\widehat{a}$ | – | – | 40 |
| Standard dev. $\sigma_b$ for $\widehat{a}$ (incl. schedule) | – | – | 0.05 |
| Temperature param. $\tau$ (incl. schedule) | – | – | 0.1 |
| No. samples for $\mathcal{L}_\Omega(\theta; \widehat{a})$ | 20 | – | 20 |
| Standard dev. $\varepsilon$ for $\mathcal{L}_\Omega(\theta; \widehat{a})$ | 0.01 | – | 0.01 |

which only know the features $v$. An episode has 100 time steps, we use 100 train, validation, and test-episodes.

**Expert policy**    Given the endogeneity of the environment, finding a globally optimal policy is computationally infeasible. We thus again resort to using a myopically optimal policy as $\bar{\pi}(s)$. Using full knowledge of $v$ and $\Phi^c$, we estimate the immediate cell costs $c_{i,j} = (v_{i,j}^c)^T \Phi^c$ for all $(i,j)$. We then apply Dijkstra's algorithm on these costs to generate the shortest path from the robot's current position to its target position [Dijkstra, 1959].

**Greedy policy**    The greedy policy estimates a straight path from the robot's current position to its target position. Disregarding cell features $v$, this is a reasonable estimate for the lowest-cost path that the robot can take.

**COAML-pipeline**    The actor model $\varphi_\omega$ is a linear model with input dimension 7 and output dimension 1. It takes $v_{i,j}$ and the time step relative to the maximum episode length as input and outputs a score $\theta$. Via a negative absolute activation, $\varphi_\omega$ ensures that all $\theta \leq 0$. We perform one pass per cell $(i,j)$. We then apply Dijkstra's algorithm on these $\theta$ to generate the best path from the robot's current position to its target position given $\theta$ [Dijkstra, 1959].

**Critic specification**    We employ double Q-learning to mitigate the critic overestimation bias in this endogenous dynamic environment [cf., van Hasselt, 2010, Fujimoto et al., 2018]. Both critics $\psi_{\beta,k}, k \in (1, 2)$ have the same structure and learning paradigm, but are initialized using different random seeds. We estimate $Q^\psi(s,a) = \frac{Q_{\psi_{\beta,1}}(s,a) + Q_{\psi_{\beta,2}}(s,a)}{2}$. The critics $\psi_{\beta,k}$ are linear models with input dimension 8 and output dimension 1. For each cell $(i,j) \in a$, they take $v$, the time step relative to the maximum episode length, and the current cost parameter $\rho$ as input, while all inputs are set to zero for $(i,j) \notin a$. After estimating a value for each cell, these values are summed to calculate $Q_{\psi_{\beta,k}}(s,a)$. We train both critics by minimizing the Huber loss between $Q_{\psi_{\beta,k}}(s_t, a_t)$ and $y_t = r_t + \gamma \, Q_{\psi_{\beta,k}}(s_{t+1}, \pi(a_{t+1}))$. For this, we use the same transitions as immediately previously for the policy update, which we sample from the replay buffer.

**Hyperparameters**    We present the hyperparameters utilized for the GSPP in Table 8

# E    Additional results

## E.1    Results for static environments

**Environment description**    The static environments, adopted from Dalle et al. [2022], are as follows: i) the Warcraft Shortest Paths Problem (WSPP) [Vlastelica et al., 2020], where the task is to compute lowest-cost paths on a map, given the raw map image as input; ii) the Single Machine Scheduling Problem (SMSP) [Parmentier and T'Kindt, 2023], where the task is to determine a job sequence for a single machine that minimizes total completion time; and iii) the Stochastic Vehicle Scheduling Problem (SVSP) [Parmentier, 2022], where the task is to find vehicle routes that minimize random delays while servicing spatio-temporally distributed tasks.

We adopt the same experimental setup as for the dynamic environments. Expert solutions are computed by solving the problems to optimality. In the static environments, we do not use critics; instead, we assume access to a black-box cost function for both the WSPP and the SMSP, and apply sample average approximation to estimate costs in the SVSP.

Figure 7, Figure 8, and Table 9 present results for the static environments. These results are generally similar to the results for the dynamic environments. In all environments, SRL performs comparably to SIL, with both algorithms approaching the expert policy. SRL consistently outperforms PPO by at least 5% and up to 54%, which converges to near-greedy policies in the WSPP and the SMSP; and which struggles to converge in the SVSP, yielding low average performance. SRL and SIL algorithms converge fast in the SVSP, requiring fewer than 10 episodes to reach a performance plateau. Both algorithms converge slower, but with the same speed, in the WSPP. These results highlight the similarity between SRL and SIL in static environments when SRL is able to sufficiently explore the combinatorial action space. They underscore that SRL is competitive with SIL and serves as a strong

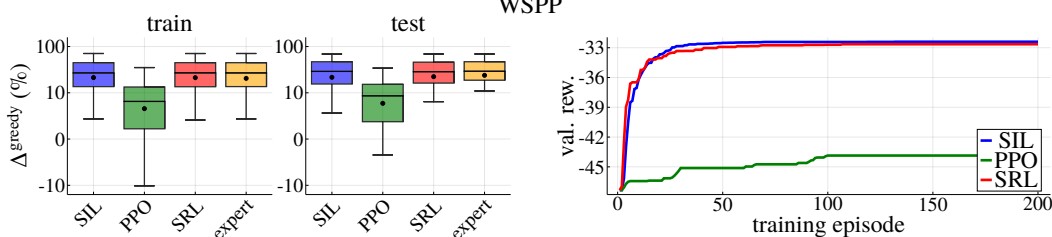

Figure 7: WSPP results. Left: final train and test-performance compared to greedy ($\Delta^{\text{greedy}}$); right: validation performance during training; averaged over 10 random model initializations.

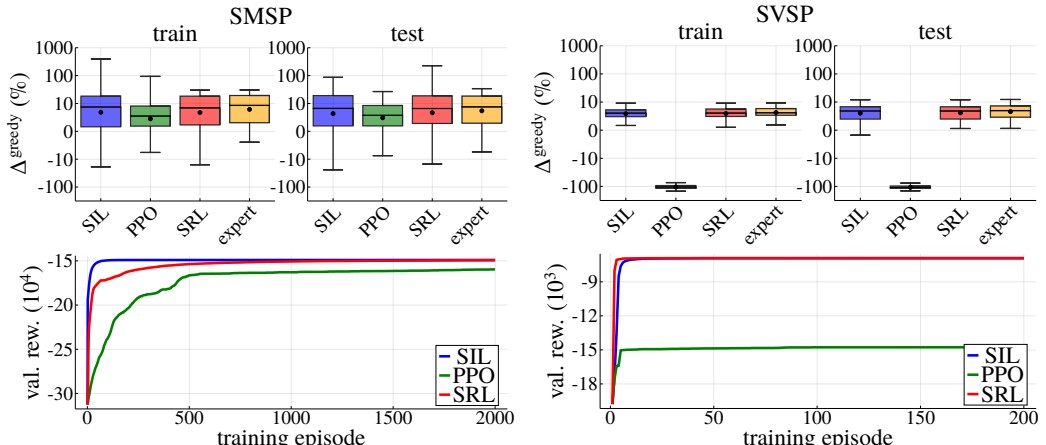

Figure 8: SMSP and SVSP results. Left: final train and test-performance compared to greedy ($\Delta^{\text{greedy}}$); right: validation performance during training; averaged over 10 random model initializations.

alternative in settings where an optimal solution is unavailable. In the SMSP, SRL and PPO converge slower than SIL, requiring 500 episodes compared to 100.

Stability measures underline the above results: SRL and SIL show low variance, while PPO is up to $400\times$ less stable. This underscores the stabilizing effect of structured learning using Fenchel-Young losses. The stability and performance gap is especially pronounced in the SVSP, where stochasticity and highly combinatorial complexity challenge PPO, in contrast to the simpler and deterministic SMSP and WSPP. These combinatorial complexities impact computational effort: while all methods complete their training in 7-9 minutes in the WSPP and in 1015 minutes in the SMSP, PPO is over $3\times$ faster than SIL and over $7\times$ faster than SRL in the SVSP, omitting offline solution time for SIL. Again, these differences stem from CO-layer usage. While negligible in simple settings like the WSPP and the SMSP, this overhead becomes substantial in environments with complicated CO-layers like the SVSP. The WSPP presents an interesting case: although it requires the largest neural network of all considered environments, its simple CO-layer reduces performance differences between algorithms. Overall, these results suggest that SRL yields the greatest performance gains in environments with highly combinatorial structure; but these gains come at the cost of increased computational effort.

Table 9: Standard deviation of validation rewards during training and final testing rewards over 10 random model initializations; and training time of algorithms in the WSPP, SMSP, and SVSP.

| Algorithm | WSPP | | | SMSP | | | SVSP | | |
|---|---|---|---|---|---|---|---|---|---|
| | train | test | time | train | test | time | train | test | time |
| SIL | 0.2 | 0.6 | 7m | 0.0 | 0.0 | 10m | 0.2 | 0.0 | 11m |
| PPO | 3.8 | 5.6 | 9m | 1.9 | 0.3 | 15m | 10.0 | 10.0 | 3m |
| SRL | 0.5 | 1.0 | 9m | 0.1 | 0.0 | 12m | 0.1 | 0.0 | 23m |

### E.2 Numerical results for all environments

We display all numerical results of the final tests in Table 10 for the static environments and in Table 11 for the dynamic environments. In all environments except the DAP, costs are to be minimized; the rewards are therefore negative. A higher number, indicating a lower cost, is therefore better. In the DAP, revenues are to be maximized; the rewards are therefore positive. A higher number, indicating higher revenues, is therefore better.

Table 10: Final performance of algorithms on the train and test dataset for the WSPP, SMSP, and SVSP. For SIL, PPO, SRL, averaged over 10 random model initializations.

| Algorithm | WSPP | | SMSP | | SVSP | |
|---|---|---|---|---|---|---|
| | train | test | train | test | train | test |
| Expert | -30.4 | -29.8 | -157306 | -152670 | -6885 | -6670 |
| Greedy | -43.1 | -43.5 | -175185 | -168738 | -7228 | -7038 |
| SIL | -30.4 | -30.5 | -159258 | -154399 | -6907 | -6701 |
| PPO | -39.3 | -39.1 | -168790 | -162892 | -14735 | -14610 |
| SRL | -30.5 | -30.6 | -159135 | -154388 | -6904 | -6695 |

Table 11: Final performance of algorithms on the train and test dataset for the DVSP, DAP, and GSPP. For SIL, PPO, SRL, averaged over 10 random model initializations.

| Algorithm | DVSP | | DAP | | GSPP | |
|---|---|---|---|---|---|---|
| | train | test | train | test | train | test |
| Expert | -30.1 | -25.5 | 569.9 | 583.2 | -1293.6 | -1284.9 |
| Greedy | -35.1 | -30.0 | 439.6 | 484.1 | -1554.2 | -1574.4 |
| SIL | -31.8 | -27.2 | 490.9 | 519.2 | -1257.7 | -1253.5 |
| PPO | -37.4 | -32.5 | 308.7 | 313.1 | -3605.0 | -3683.9 |
| SRL | -31.9 | -27.3 | 529.8 | 555.3 | -275.5 | -280.2 |

