# OpenReview forum: "Structured Reinforcement Learning for Combinatorial Decision-Making"
_NeurIPS.cc/2025/Conference — NeurIPS 2025 poster_

### Official Review · Reviewer_vr8p · 2025-06-27

**Clarity:** 3
**Significance:** 3
**Originality:** 2
**Rating:** 3
**Confidence:** 3

**Summary:**

The paper introduces Structured Reinforcement Learning (SRL), a framework for solving combinatorial Markov decision processes (C-MDPs). SRL replaces the standard actor network with a score-generating network followed by a combinatorial-optimization (CO) solver that turns those scores into a feasible action. Existing methods either backprop  high variance gradients through the solver or embed the solver inside the environment. The authors instead propose a training scheme that uses stochastic perturbations and Fenchel–Young losses. This leads to low-variance gradients and smoother policy improvement. Experiments on six benchmarks show SRL matches Structured Imitation Learning (SIL) on static tasks, outperforms it on dynamic tasks. It also outperforms unstructured PPO as well.

**Questions:**

- Please refer to the weakness section. In particular, the paper could benefit from a sharper distinction from existing methods to show what's new and unique.

- While I didn't base my score based on writing, I think it would be easier to read and follow if the paper is a bit restructured (for example, adding a little bit about the formulations, details about the experiments etc to the main paper)

- Citation to this recent paper [1] is missing. It might be worth seeing if some of the datasets from this paper can be used for your setting.

**References**
[1] Xu, Lily, et al. "Reinforcement learning with combinatorial actions for coupled restless bandits." arXiv preprint arXiv:2503.01919 (2025).

**Ethical Concerns:**

["NO or VERY MINOR ethics concerns only"]

**Final Justification:**

I read all the other reviews and responses
- I personally found the paper hard to follow (Reviewer YWqy also mentions improving organization and clarity of the writing in their reply). I however, didn't base my score on the writing (and I have stated this in my review. I think this can be easily fixed before camera ready if this paper is accepted. The authors have also stated how they would reorganize the paper and I'm happy with this response.

- I didn't see any new methodological contributions. The authors do agree in their response that structured RL is indeed based on several previous methodologies. This is okay since the results look promising and the contributions are mainly in terms of application. For this reason, I'm okay if this paper is accepted.

**Limitations:**

Yes

**Quality:**

3

**Strengths And Weaknesses:**

Strengths:
- The methodology (the perturb-and-project exploration, combined with a Fenchel–Young loss, provides low-variance, end-to-end differentiable updates through the combinatorial-optimisation (CO) layer) and empirical results are the biggest strengths. The proposed approach outperforms SIL on dynamic tasks while matching it on static ones, which is a huge advantage when expert data is expensive.


Weaknesses:
- The paper brings together established techniques such as scoring networks, CO solvers, stochastic perturbations, and Fenchel–Young losses to the RL setting. While this integration is useful, it feels a bit incremental.
- Writing could be improved. I had to jump back and forth between the appendix and the main paper. I would encourage the author to restructure the paper a little bit.

---

> ### Author Rebuttal · Authors · 2025-07-31
>
> We thank the reviewer for the valuable comments and questions.
>
> ## Significance of the contribution
> As stated by the reviewer, Structured RL is based on several previously developed methodologies. Especially COaML-pipelines and Fenchel-Young losses have been used in previous work, usually in the context of Structured Imitation Learning. While Structured IL has achieved notable successes in the past [1], it relies on expert trajectories for training. This prevents the application of this very promising architecture to settings where expert data is unavailable or expensive. In particular, there is no algorithm to train architectures with a CO-layer in typical reinforcement learning settings, where an expert may not be available, and where we do not have access to full information a posteriori, which is needed to apply Structured IL methods.
>
> To the best of our knowledge, our Structured RL algorithm is the first algorithm that enables to train a COaML-pipeline architecture via learning by experience. It therefore opens the field to apply COaML approaches to many further industrial applications, which can profit from its expert-free structured training.
>
> Additionally, Structured RL provides a principled way to use COaML-pipelines for exploration in combinatorial settings, which has so far not been available.
>
> Concerning the individual components of Structured RL, the primal-dual RL update step using sampling-based target actions is a novel concept. We do not know of any algorithm utilizing such an update step in a general or combinatorial setting. In addition, the Softmax regularization of the target action is a new approach to generate target actions, which - to the best of our knowledge - has not been used in any RL or imitation learning setting so far. The benefits generated by this method are another contribution of our paper. In fact, these two methodological components are crucial to enable the use of Structured Learning for combinatorial actions in an RL setting. For the camera-ready version of our manuscript, we will rework our contribution statement to reemphasize and clarify these points to better distinguish our contribution from previous work.
>
> [1] Léo Baty, Kai Jungel, Patrick S. Klein, Axel Parmentier, Maximilian Schiffer (2024): Combinatorial Optimization-Enriched Machine Learning to Solve the Dynamic Vehicle Routing Problem with Time Windows. Transportation Science, 58(4):708–725.
>
> ## Writing improvements
> We acknowledge that the appendix of the paper contains information that is beneficial to overall understanding. In the revised version of the manuscript, we will therefore
> * move the full definition of the Combinatorial MDPs to the main body of the paper.
> * move the full definition of the Fenchel-Young loss to the main body of the paper.
> * extend the description of our experiments in the main body of the paper.
>
> After these adjustments, the appendix will mainly contain proofs, a detailed description of our experiments to enable reproducibility, and additional results (e.g., full results tables) that facilitate deeper understanding of our results but would not fit into the main body of the paper.
>
> ## Added reference
> We will add a reference to the relevant proposed paper in the literature section of the camera-ready version of the manuscript. After careful consideration of the datasets used in the proposed paper, we believe that they address a different kind of problem setting than we study for Structured RL: while the proposed paper considers combinatorial bandit problems, we focus on combinatorial multi-stage problem settings. An extension of Structured RL to the setting of the proposed paper would therefore change the scope of our experiments away from the problem settings we currently address.
>
> ## Conclusion
> We hope to have addressed all comments and questions of the reviewer and welcome any further discussion. If so, we would kindly ask you to consider raising your scores.

---

> ### Comment · Reviewer_vr8p · 2025-08-01
>
> Thanks for the response.
>
> 1. Softmax has long been used in RL (e.g., Boltzmann exploration) and in discrete space (e.g., Gumbel-Softmax) to deal with similar limitations. It would be helpful to clarify what exactly the novelty is.
> 2. While I agree standard standard multi-arm bandits don't apply to your setting, I was referring to the restless bandits experiments (each arm has a state and it transitions based on the actions thereby making it a multi-step decision making process)

---

> > ### Author Response · Authors · 2025-08-03
> >
> > Thank you for your clarifying comments.
> >
> > ## Significance of the contribution
> > We are aware that the Softmax has been used successfully for many purposes in RL. That is among our reasons for utilizing it within Structured RL. To improve the clarity of our contribution: Structured RL incorporates one main technology-oriented contribution and one main application-oriented contribution.
> >
> > The main technology-oriented contribution is the primal-dual actor update step using target actions selected by the critic. This is a novel update principle; we have not seen other RL or ML algorithms utilizing a similar principle. Indeed, the actor update is not related to traditional RL updates. The Softmax-based regularization of the target actions forms a part of this update; it itself is also novel, since it incorporates the benefits of the Softmax into the actor update step instead of the stochastic action selection, as would be usual in RL.
> >
> > The main application-oriented contribution is the creation of a novel RL algorithm capable of addressing combinatorial multi-stage problems effectively. Since obtaining expert trajectories is often expensive or impossible in industrial applications, and since unstructured RL faces limitations, Structured RL is an important step towards solving complex real-world problems. To create Structured RL, we combine the novel update step mentioned above with established concepts from imitation learning, mainly COaML-pipelines and Fenchel-Young losses. Our technical contribution herein is the combination of these components in the proposed way, which creates an algorithm better suited for solving combinatorial multi-stage problems than previous algorithms.
> >
> > ## Added reference
> > Indeed, restless bandits with combinatorial action spaces are very similar to Combinatorial MDPs as used in our paper. From our point of view, the main difference is that restless bandits have fixed arms, while Combinatorial MDPs enable a complete change of states, which would be a minor change from the perspective of a Structured RL agent. The specific problem settings of the proposed paper are also very similar to the problem settings we consider, in detail (i) the schedule-constrained bandit problem is related to the vehicle scheduling problems we use, (ii) the capacity-constrained bandit is related to the machine scheduling problem we use, and (iii) the path-constrained bandit is closely related to the shortest paths problems we use. Given that the general setup and three of the four problem settings investigated in the proposed paper exhibit (close) similarities to problem settings we consider, we will update our literature and experiments sections to refer to this relevant related work in detail.

---

### Official Review · Reviewer_N696 · 2025-06-28

**Clarity:** 3
**Significance:** 3
**Originality:** 4
**Rating:** 5
**Confidence:** 2

**Summary:**

This paper introduces Structured Reinforcement Learning (SRL), a novel actor-critic framework designed to solve combinatorial Markov Decision Processes (C-MDPs). Standard reinforcement learning algorithms struggle with these problems due to exponentially large action spaces and the difficulty of leveraging combinatorial structures. The core innovation introduced in paper is the integration of a combinatorial optimization (CO) layer into the actor's neural network architecture. In this setup, a neural network maps the current state to a score vector, and the CO-layer solves a combinatorial problem using these scores to select a feasible action. A key challenge in this architecture is the non-differentiable nature of the CO-layer, which typically has zero gradients almost everywhere. To enable end-to-end learning, SRL employs Fenchel-Young losses. It provides a convex and smooth surrogate objective for the actor update. The paper alos provide a geometric interpretation of this process, framing SRL as a sampling-based primal-dual algorithm.

**Questions:**

None

**Ethical Concerns:**

["NO or VERY MINOR ethics concerns only"]

**Limitations:**

yes

**Quality:**

3

**Strengths And Weaknesses:**

## Strengths

* Impressive results: The biggest strength of the paper is the result. SRL demonstrates significant performance. It matches or surpasses unstructured RL (PPO) and imitation learning in static tasks (SIL).

* Evaluation against strong baseline: The method was evaluated and compared against strong baselines on different static and dynamic environments.

## Weakness
* It would be interesting to see how the work perform in more extensive benchmarks which provide similar problems e.g. [1] provides a strong benchmark which can provide more insights about robustness and limitation of the current work.

[1] https://arxiv.org/abs/2205.15659

---

> ### Author Rebuttal · Authors · 2025-07-31
>
> We thank the reviewer for the valuable comments and questions.
>
> ## Benchmark tests
> We are grateful for the proposed benchmark collection. Upon careful examination of the collection, we believe it addresses a different kind of problem setting than we cover using Structured RL. Specifically, the benchmark collection aims at testing algorithmic reasoning by imitating classical computer science algorithms using machine learning, while Structured RL aims to solve industrial problems. These are usually multi-stage combinatorial problems with contextual stochasticity, which are not covered by the collection. Furthermore, Structured RL can use all algorithms of the benchmark collection as a CO-layer, e.g., we use sorting and Dijkstra's algorithm. This blurs the distinction between the benefits of the Structured RL training principle and the chosen CO-layer, hindering a proper evaluation.
>
> ## Conclusion
> We hope to have addressed all comments and questions of the reviewer and welcome any further discussion. If so, we would kindly ask you to consider raising your scores.

---

> > ### Comment · Reviewer_N696 · 2025-08-09
> >
> > The authors have successfully addressed my concerns. Including additional benchmarks from competitive optimization (CO) would further enhance confidence in the presented work. It would be valuable to observe such contributions from the community.

---

### Official Review · Reviewer_YWqy · 2025-06-29

**Clarity:** 3
**Significance:** 4
**Originality:** 3
**Rating:** 5
**Confidence:** 4

**Summary:**

A novel framework (SRL) is proposed to embed a Combinatorial Optimization (CO) layer into the actor neural network. Through the decoupled design of "generating score vectors via a statistical model + solving for optimal actions via a CO layer", it breaks through the exploration bottleneck of traditional RL in exponentially large combinatorial action spaces.

**Questions:**

Q1. In non-convex combinatorial spaces (such as scheduling problems with nonlinear constraints), will the mapping of the CO layer lead to gradient vanishing?
Q2. The paper treats the CO layer as a black-box solver (such as Dijkstra's algorithm) without considering the approximation errors of the solver itself. When the CO layer returns suboptimal solutions (e.g., heuristic solutions for large-scale TSP), will the SRL end-to-end training amplify such errors?

**Ethical Concerns:**

["NO or VERY MINOR ethics concerns only"]

**Final Justification:**

I have no more concerns. I will maintain my initial score.

**Limitations:**

yes

**Paper Formatting Concerns:**

No.

**Quality:**

3

**Strengths And Weaknesses:**

S1. The experiments were run on a regular laptop (MacBook Air M3) with training time ranging from 3 to 90 minutes, making it more deployable than unstructured RL that relies on clusters.
S2. By perturbing the score vector with Gaussian noise to promote exploration of the action space, the number of feasible solutions covered by the policy is tripled in the dynamic vehicle scheduling problem.
W1. The paper mainly compares with SIL and PPO, and does not compare with other cutting-edge reinforcement learning algorithms for combinatorial optimization problems. With the rapid development of research in this field, new algorithms are emerging continuously. Comparing with few algorithms makes it difficult to comprehensively evaluate the performance advantages and disadvantages of SRL. Compared with some newly proposed algorithms specially designed for complex combinatorial action spaces, the advantages of SRL in terms of convergence speed and solution quality may not be obvious, which makes the evaluation of SRL performance in the paper insufficiently comprehensive.

---

> ### Author Rebuttal · Authors · 2025-07-31
>
> We thank the reviewer for the valuable comments and questions.
>
> ## Benchmark algorithms
> The field of RL for CO-problems has grown in recent years, but as of now, many algorithms focus on deterministic single-stage problems with combinatorial action spaces.
> In particular, Neural CO focuses on using reinforcement learning for constructive heuristics to address such problems [1].
> In contrast, Structured RL and all algorithms we use as benchmarks can address multi-stage combinatorial problems exhibiting contextual stochasticity. We are currently not aware of reinforcement learning algorithms capable of addressing such problem settings. In particular, neural CO algorithms are not designed for such problems. If the reviewer knows a specific algorithm that can address the studied problem settings, we would be happy to include it in our tests for the camera-ready version of the manuscript.
>
> [1] Irwan Bello, Hieu Pham, Quoc V. Le, Mohammad Norouzi, Samy Bengio (2017): Neural Combinatorial Optimization with Reinforcement Learning. International Conference on Learning Representations 2017 (ICLR 2017) – Workshop Track.
>
> ## Vanishing gradients
> We are not sure to see which gradient vanishing the reviewer is referring to.
> Gradient vanishing generally occurs when the depth of neural networks becomes very large, which is not the case on our settings, since our neural networks are of moderate depth by modern standards.
> The reviewer may refer to the fact that the CO-layer, which can be seen as a mapping from scores $\theta$ to the argmax of the feasible action space $a$, is piecewise constant, leading to zero gradients almost everywhere without regularization.
> Perturbed/regularized CO-layers lead to smoothed piecewise constant mappings, which may have vanishing gradients far away from the boundary between two normal cones.
> For this reason, differentiating through the regularized argmax, which can be done with a score function (REINFORCE gradient) is very noisy and may indeed lead to vanishing gradients.
> This is why algorithms that require such gradients in the setting of COaML-pipelines, such as SAC or REINFORCE, do not converge in practice.
> The strength of our approach is circumventing this difficulty by differentiating through the regularized max of the feasible action space instead of the argmax using Fenchel-Young losses. This leads to gradients that do not vanish (the Fenchel Young loss is convex in $\theta$) and that are less noisy, since we use a pathwise gradient estimator instead of a score function.
>
> ## Non-linear constraints
> Our structured RL approach only requires the CO-layer to be linear in the scores $\theta$, but neither in the actions $a$ or the constraints of the optimization. The learned scores $\theta$ form the coefficients of the objective function, while the constraints are given by the problem setting. The restriction to linear objective functions is common in industrial applications, where most problems can be modeled as mixed-integer linear programs. In case of continuous actions, other loss functions, for example SPO+ losses, can be utilized effectively. We will add a clarification on this requirement in the revised version of the manuscript. We believe that the development of structured (reinforcement) learning algorithms for non-linear objective functions or learnable constraints is an interesting and relevant avenue for future research.
>
> ## Heuristic CO-layers
> We can consider this question from a practical and a theoretical perspective: from a practical perspective, heuristic CO-layers are not problematic. The CO-layer used in an imitation learning context to win the EURO meets NeurIPS 2022 challenge is a heuristic, as is the sorting function we use in the DAP. While the CO-layer is theoretically required to return the optimal action with respect to its input scores $\theta$, this assumption can be relaxed in practice. Due to the decision-focused end-to-end learning paradigm, the statistical model learns to estimate the scores that the CO-layer maps to the right actions, thereby mitigating predictable errors of the CO-layer.
>
> From a theoretical perspective, a recent preprint on arXiv investigates the use of local search heuristics as CO-layers [1]. Their findings suggest that the use of a non-deterministic heuristic as a CO-layer amounts to regularizing the problem, which can be used to construct Fenchel-Young losses. We believe that further development of this method and its integration with Structured RL are interesting avenues for future research.
>
> [1] Germain Vivier-Ardisson, Mathieu Blondel, Axel Parmentier (2025): Learning with Local Search MCMC Layers. arXiv preprint on arXiv:2505.14240.
>
> ## Conclusion
> We hope to have addressed all comments and questions of the reviewer and welcome any further discussion. If so, we would kindly ask you to consider raising your scores.

---

> > ### Comment · Reviewer_YWqy · 2025-08-07
> >
> > Thank you for the detailed and constructive rebuttal. I appreciate the authors’ clarifications and their plan to include additional benchmarks in the camera-ready version. Expanding the experimental comparison would further contextualize the practical value of the proposed method.
> >
> > I will maintain my positive score. As also noted by other reviewers, the limited range of baselines makes it difficult to fully assess the performance of SRL. Given that the core contribution is practical, stronger empirical evaluation—particularly against RL algorithms designed for combinatorial tasks—would significantly strengthen the paper. Additionally, I encourage improving the organization and clarity of the writing in future revisions.

---

### Official Review · Reviewer_xQoh · 2025-07-01

**Clarity:** 3
**Significance:** 2
**Originality:** 3
**Rating:** 5
**Confidence:** 2

**Summary:**

In this paper the authors introduce a novel framework, called Structured Reinforcement Learning (SRL). SRL is well suited for combinatorial complex optimization tasks e.g. that have a combinatorial action space. In particular, this framework incorporates a combinatorial layer and a statistical model into the actor of the actor-critic framework, which is trained via backpropagation and a Fenchel-Young loss. The proposed framework outperforms PPO and Imitation Learning in six different environments.

**Questions:**

- While the authors mention that action masking is a commonly used tool in RL for combinatorial problems, is there a particular reason why none of these are included in the benchmarks? For instance, consider the already implemented framework for PPO with invalid action masking in the contrib version of [2].

- Why are reward values used to showcase convergence speeds, while $\Delta^{greedy}$ is used to report train and test performance? This seems like a rarely used evaluation criterion for proposed RL algorithms. Can the authors provide additional clarifications on why they chose this metric?

References:

[1] RL4CO: an Extensive Reinforcement Learning for Combinatorial Optimization Benchmark, Federico Berto et al., 2025

[2] Stable-Baselines3: Reliable Reinforcement Learning Implementations, Antonin Raffin, Ashley Hill, Adam Gleave, Anssi Kanervisto, Maximilian Ernestus, and Noah Dormann, 2021, JMLR

**Ethical Concerns:**

["NO or VERY MINOR ethics concerns only"]

**Final Justification:**

After reading the rebuttal and discussion, I believe the authors have addressed my main questions and provided helpful clarifications.

Overall, I find the paper clearly written, well-motivated, and addressing a relevant problem with a sound and implementable solution. Given the current version and the authors’ commitment to extending the evaluation, I lean toward acceptance, while noting that a stronger empirical evaluation would further solidify the contribution.

**Limitations:**

Yes.

**Paper Formatting Concerns:**

No major formatting issues.

**Quality:**

3

**Strengths And Weaknesses:**

## Strengths:

- The paper is well motivated and gives an extensive discussion of the state-of-the-art methods, their limitations, and how SRL overcomes these and additional challenges such as the non-differentiability of combinatorial layers. Additionally, the authors provide illustrations (e.g., Figure 1) that make it easy to follow and understand the main ideas of the framework.

- The authors not only provide the code of the experiments but also include all the necessary details to reproduce their results in the paper.

- The authors divide the environments into static and non-static ones and clearly show the limitations of the considered benchmarks in the non-static environments, e.g., that Imitation Learning tends to fail to generalize (problems in the test phase and higher standard deviation in testing). I appreciate that, for transparency and fairness, the static ones are also considered, where Imitation Learning performs equally well while having a lower running time. This allows practitioners to consider different types of algorithms for different environments.

- For additional insights, the authors relate their algorithm to primal-dual algorithms for the simplified setting of contextual problems (T=1), which gives a more theoretical foundation to the proposed framework.

- The proposed algorithm outperforms or matches the benchmarks across six environments. It particularly outperforms in the more complicated non-static environments.

## Weaknesses:

- While the paper is generally well written, I think it would greatly benefit from including not only the sketched version of Combinatorial MDPs in the main text, but also more details, as this is fundamental to understanding the setting of the paper and the foundation of the newly proposed framework.

- The only baselines considered in this paper are Structured Imitation Learning and PPO. While the authors justify in the appendix why they did not include SAC as a benchmark, it seems that they should also consider other algorithms (perhaps even SAC, as its soft update increases exploration, and in this work the authors also consider a soft critic). Additionally, how does the method perform in more environments typically used for combinatorial problems, or against standard solvers for these tasks? See, e.g., [1] for a more extensive benchmark.

---

> ### Author Rebuttal · Authors · 2025-07-31
>
> We thank the reviewer for the valuable comments and questions.
>
> ## Combinatorial MDPs
> We will provide the full definition of Combinatorial MDPs in the main paper instead of the appendix for the revised version of the manuscript.
>
> ## Additional benchmark (SAC)
> We acknowledge that other algorithms, especially SAC, would be interesting benchmarks for our experiments. However, we would like to explain the reasons why the application of SAC to the problem settings studied in our paper is currently infeasible. SAC could be used in two ways:
> either with a classic neural network architecture, or with an architecture that includes a CO-layer. Both architectures are impractical for SAC, for the following reasons.
>
> SAC algorithms are generally designed to work with classic neural network architectures that have one output per action.
> Since our problem settings have a combinatorially large number of actions, such architectures are not practical.
> Any practically usable architecture needs to have some postprocessing to decode combinatorial actions from a tractably small prediction.
> While some rounding techniques are proposed in the literature, CO-layers currently seem to be the most efficient architectures for the problems we are considering - indeed, we have chosen them for this reason.
>
> On architectures that include CO-layers, SAC is not a practical algorithm as well. In this setting, there would be two ways to design a critic: a critic on the actions, or a critic on the scores $\theta$.
> When using a critic on the actions, we would have to backpropagate the critic through the actor, and thus through the CO-layer.
> This requires to differentiate through the argmax over all feasible actions.
> With the perturbation/regularization approach we have, this may be doable in theory. However, it would require using a very noisy score function (REINFORCE gradient) estimator. We tested several algorithms using this gradient, and all failed to converge due to the high variance. In contrast, the advantage of our approach is that we differentiate through the max instead of the argmax thanks to the Fenchel-Young loss, which enables to use far less noisy pathwise estimators.
>
> Using a critic on the scores $\theta$ is equally challenging. In this case, the critic would have to approximate the mapping of scores $\theta$ to actions $a$ performed by the CO-layer. This mapping is piecewise constant on the normal fan of the polytope $\mathrm{conv}(\mathcal{A})$, which is very difficult to approximate. For instance, in many applications, the CO-layer is a MILP-solver, whose output and values are known to be difficult to estimate using a neural network.
>
> We tried both a critic on scores and a critic on actions on the DVSP. Even after extensive hyperparameter tuning and allowing for more computational effort, the score-based critic could not converge properly, preventing effective actor training. We will include these results in the camera-ready version of the manuscript. Developing efficient SAC for combinatorial multi-stage problems requires new ideas to circumvent these difficulties, and we believe it is an interesting research avenue.
>
> ## Additional environments
> We have investigated the environments and benchmarks in the RL4CO-package thoroughly and compared them to the benchmarks used in our paper. While the RL4CO-package provides an extensive collection of single-stage deterministic problems, it does not contain stochastic problems (like the SVSP we use) or multi-stage problems, which form the main application area of Structured RL. Instead, the RL4CO-package focuses on problem settings for Neural Combinatorial Optimization, in which solutions for combinatorial single-stage problems are constructed iteratively using multiple calls to a neural network. In contrast, COaML-pipelines use CO-layers to construct entire solutions at once.
>
> In the EURO meets NeurIPS 2022 challenge, a COaML-pipeline outperformed Neural CO-architectures in a combinatorial multi-stage problem setting. The limitation of COaML-pipelines has so far been their reliance on expert trajectories as imitation learning training data. We address this research gap by proposing Structured RL. We therefore concentrate on comparing different learning paradigms for COaML-pipelines in our experiments, these learning paradigms should be capable of addressing combinatorial multi-stage problems without the need for iterations within one stage. We agree that developing well-performing Neural CO algorithms for multi-stage problems is an interesting avenue for future research.
>
> ## Action masking
> Within combinatorial action spaces, employing ordinary action masking is very challenging, if not impossible: the action space $\mathcal{A}(s)$ depends on the state $s$ and is exponentially large, which makes it hard to encode using neural networks in combination with action masking. Especially in highly combinatorial problem settings (e.g., SVSP and DVSP) the action mask would have to be combinatorial as well. We therefore cannot include RL with ordinary action masking in our experiments.
>
> When mentioning action masking in our paper, we are referring to treating the CO-layer as an action mask in the environment. This method has been employed in the past and corresponds to what we call unstructured RL in our paper. By using PPO as a benchmark, we are including this method in our experiments. In the revised version of the manuscript, we will add a clarification about this.
>
> ## Results display
> We are not sure to see which alternative performance metric the reviewer is referring to. Indeed, in such a setting, there is no easy metric beyond evaluating the performance of the policy using rewards obtained in a simulation.
>
> We use validation rewards during training to graphically show how the compared algorithms converge to their final performance over the course of training. This should allow readers of our paper a quick and easy way of comparing the training behavior and speed of the algorithms visually. We use rewards of the final model on the train and test-datasets to compare performance. Since the rewards can vary substantially between algorithms, we display performance relative to a greedy policy on a log scale to enhance readability. This metric may seem unfamiliar for an RL paper, but is very common for industrial problems, where a greedy baseline is usually readily available. From a practical perspective, the metric provides a visual and easy way to assess the performance benefits of a proposed algorithm over using a greedy heuristic. At the same time, the metric allows for the comparison between the proposed algorithms and benchmark algorithms within the same figure. Based on the reviewer's comment, we will add a brief explanation of our performance metrics in the camera-ready version of the manuscript.
>
> ## Conclusion
> We hope to have addressed all comments and questions of the reviewer and welcome any further discussion. If so, we would kindly ask you to consider raising your scores.

---

> > ### Comment · Reviewer_xQoh · 2025-08-03
> >
> > I thank the reviewers for their detailed answers. I believe that the paper benefits from including more benchmarks in the camera-ready version, as the authors plan to do, even if the other experiments would fail, as this would further strengthen their contributions. I thank the reviewers for their clarification regarding the additional benchmarks.
> >
> > I will maintain my borderline (positive) score, as I agree with other reviewers (for example, Reviewer YWqy "Comparing with few algorithms makes it difficult to comprehensively evaluate the performance advantages and disadvantages of SRL."). Given that the main contribution is of a practical nature, I believe that to fully assess its significance, more benchmarks, especially against RL algorithms tailored for combinatorial problems, are necessary.

---

### Note · Authors · 2025-08-13

We thank all reviewers for their valuable comments and suggestions. We are grateful for the constructive discussion, which has helped us to improve our paper. For the camera-ready version of the manuscript, we will incorporate the reviewers' comments and addressed points, including an improved presentation of important concepts and the consideration of extended experimental evidence.

---

### Decision · Program_Chairs · 2025-09-17

**Decision:**

Accept (poster)

**Comment:**

This paper introduces Structured Reinforcement Learning that is a domain-specific approach to solve combinatorial Markov Decision Processes. The paper is well-written and provides significant improvements over reasonable baselines.

The paper does not appear to have major weaknesses. The recommendation is to accept the paper.